# RNF41 regulates the damage recognition receptor Clec9A and antigen cross-presentation in mouse dendritic cells

Kirsteen M Tullett[1†], Peck Szee Tan[1†], Hae-Young Park[1], Ralf B Schittenhelm[2], Nicole Michael[1], Rong Li[3], Antonia N Policheni[4,5], Emily Gruber[1], Cheng Huang[2], Alex J Fulcher[6], Jillian C Danne[7], Peter E Czabotar[4,5], Linda M Wakim[8], Justine D Mintern[9], Georg Ramm[1,7], Kristen J Radford[10], Irina Caminschi[1,8], Meredith O'Keeffe[1], Jose A Villadangos[8,9], Mark D Wright[11], Marnie E Blewitt[4,5], William R Heath[8], Ken Shortman[4,5], Anthony W Purcell[1], Nicos A Nicola[4,5], Jian-Guo Zhang[4,5], Mireille H Lahoud[1]*

[1]Infection and Immunity Program, Monash Biomedicine Discovery Institute and Department of Biochemistry and Molecular Biology, Monash University, Clayton, Australia; [2]Monash Proteomics and Metabolomics Facility, Monash Biomedicine Discovery Institute and Department of Biochemistry and Molecular Biology, Monash University, Clayton, Australia; [3]Centre for Biomedical Research, Burnet Institute, Melbourne, Australia; [4]The Walter and Eliza Hall Institute of Medical Research, Parkville, Australia; [5]Department of Medical Biology, University of Melbourne, Parkville, Australia; [6]Monash Micro Imaging Facility, Monash University, Clayton, Australia; [7]Ramaciotti Centre for Cryo-Electron Microscopy, Monash University, Clayton, Australia; [8]Department of Microbiology and Immunology at the Peter Doherty Institute for Infection and Immunity, The University of Melbourne, Melbourne, Australia; [9]Department of Biochemistry and Molecular Biology at the Bio21 Molecular Science and Biotechnology Institute, The University of Melbourne, Melbourne, Australia; [10]Mater Research Institute - University of Queensland, Translational Research Institute, Brisbane, Australia; [11]Department of Immunology, Monash University, Melbourne, Australia

*For correspondence:
mireille.lahoud@monash.edu

[†]These authors contributed equally to this work

**Abstract** The dendritic cell receptor Clec9A facilitates processing of dead cell-derived antigens for cross-presentation and the induction of effective CD8[+] T cell immune responses. Here, we show that this process is regulated by E3 ubiquitin ligase RNF41 and define a new ubiquitin-mediated mechanism for regulation of Clec9A, reflecting the unique properties of Clec9A as a receptor specialized for delivery of antigens for cross-presentation. We reveal RNF41 is a negative regulator of Clec9A and the cross-presentation of dead cell-derived antigens by mouse dendritic cells. Intriguingly, RNF41 regulates the downstream fate of Clec9A by directly binding and ubiquitinating the extracellular domains of Clec9A. At steady-state, RNF41 ubiquitination of Clec9A facilitates interactions with ER-associated proteins and degradation machinery to control Clec9A levels. However, Clec9A interactions are altered following dead cell uptake to favor antigen presentation. These findings provide important insights into antigen cross-presentation and have implications for development of approaches to modulate immune responses.

## Introduction

Damage-associated molecular patterns (DAMPs) are endogenous molecules that are normally sequestered in healthy cells, but are released or exposed upon injury or infection and thereby signal 'danger' to the immune system. DAMPs include a wide variety of molecules including nucleic acids, heat shock proteins, HMGB1, ATP, and mediators produced during death receptor signaling (*Vénéreau et al., 2015*; *Yatim et al., 2015*). The recognition of DAMPs by pattern recognition receptors on dendritic cells (DC) shapes immune responses by influencing their capacity to take up, process and present antigen (Ag) to T cells for induction of immunity or tolerance (*Alloatti et al., 2016*; *Brown, 2012*).

Individual DC subsets have unique expression patterns of receptors that enable them to identify particular DAMPs. The CD8$\alpha^+$ DC lineage in mice and the equivalent CD141$^+$ DC lineage in humans, now both termed cDC1 (*Guilliams et al., 2014*), are particularly adept at cross-presentation: the uptake and processing of antigenic material from dead cells for presentation on MHC class I molecules, a process critical for immunity to intracellular pathogens and cancer (reviewed in *Radford et al., 2014*). The C-type lectin-like receptor Clec9A (also known as DNGR1) is critical for recognition and processing of dead cells, and is a key defining marker of the cDC1 subset (*Ahrens et al., 2012*; *Caminschi et al., 2008*; *Huysamen et al., 2008*; *Sancho et al., 2009*; *Sancho et al., 2008*; *Zhang et al., 2012*).

Clec9A recognizes dead cells once their cell membrane is disrupted, exposing internal cellular components (*Ahrens et al., 2012*; *Sancho et al., 2009*; *Zhang et al., 2012*). The Clec9A ligand exposed upon cell death is the filamentous form of actin (F-actin), which is normally associated with actin-binding proteins (*Ahrens et al., 2012*; *Schulz et al., 2018*; *Zhang et al., 2012*) and potentially with Ag from within the dead cell. Although Clec9A engagement can facilitate signaling via the tyrosine kinase Syk (*Schulz et al., 2018*; *Zelenay et al., 2012*), Clec9A engagement by dead cell ligands does not appear to directly activate the DC; rather, its role appears to be in regulating the trafficking and processing of the dead cell-derived material once it enters the DC (*Zelenay et al., 2012*). Clec9A directs material to early and recycling endosomes, favoring Ag processing and presentation, rather than to phagolysosomes for degradation (*Zelenay et al., 2012*). cDC1 from Clec9A knockout mice have a reduced capacity for cross-presentation of dead cell-derived Ag (*Sancho et al., 2009*) and are more susceptible to viral infection (*Zelenay et al., 2012*). Clec9A therefore appears to direct Ag to the MHC class I presentation pathway to initiate anti-viral CD8$^+$ T cell immune responses.

Consistent with the role of Clec9A in enhancing immunity, delivering Ag to cDC1 using monoclonal antibodies against Clec9A, has proved very effective in enhancing immune responses and improving vaccine efficacies (*Caminschi et al., 2008*; *Lahoud et al., 2011*; *Sancho et al., 2008*; *Tullett et al., 2016*). Targeting Clec9A not only enhances CD8$^+$ T cell responses and cytotoxic T cell generation (*Caminschi et al., 2008*; *Sancho et al., 2008*), but surprisingly also promotes CD4$^+$ T cell responses, as supported by the role of cDC1 in promoting both CD8$^+$ and CD4$^+$ T cell responses (*Ferris et al., 2020*). Indeed, targeting Clec9A primes follicular helper T cells and induces potent humoral responses even in the absence of adjuvants (*Caminschi et al., 2008*; *Lahoud et al., 2011*). The capacity of Clec9A to promote both MHC class I and MHC class II presentation prompted a closer examination of the mechanisms underpinning Clec9A handling of Ag, and a search for intracellular interacting proteins that may play a role in the regulation of Clec9A function.

In this study, we identified a novel interaction between Clec9A and the E3 ubiquitin ligase RNF41 (also known as Nrdp1/FLRF). RNF41, a member of the RING family widely expressed in immune cells, directly binds and ubiquitinates target proteins, targeting them for downstream fates such as proteasomal degradation (e.g. ErbB3, MyD88, USP8) or to regulate protein trafficking (VPS52) (*Bouyain and Leahy, 2007*; *De Ceuninck et al., 2013*; *Masschaele et al., 2017*; *Wang et al., 2009*). RNF41 also modulates immune responses through Toll-like receptor pathways by ubiquitinating MyD88 and the kinase TBK1 (*Wang et al., 2009*). In addition, RNF41 regulates the sorting of endosomal proteins by ubiquitinating Ubiquitin-Specific Protease 8 (USP8), which in turn regulates the stability of the Endosomal Sorting Complexes Required for Transport (ESCRT)−0 protein complex (*De Ceuninck et al., 2013*). Accordingly, we investigated the molecular basis and the intracellular consequences of the Clec9A-RNF41 interaction. Here we report that RNF41 regulates Clec9A via a novel mechanism. E3 ubiquitin ligases usually regulate the quantity and quality of cell surface receptors via an interaction with the cytosolic, transmembrane or membrane proximal regions of the

target receptor (*Diamonti et al., 2002*; *Tan et al., 2019*). Intriguingly, we show that RNF41 interacts with and ubiquitinates the extracellular domains of Clec9A within the cell, to regulate Clec9A expression and function. We demonstrate that RNF41 colocalizes with the extracellular domain of Clec9A within cross-presenting DC, and acts as a negative regulator of Clec9A and hence cross-presentation by cDC1. Our results highlight a novel pathway for the regulation of Clec9A receptor fate, revealing further novel Clec9A interactions and providing insights into the mechanisms underpinning DAMP recognition and Ag cross-presentation.

## Results

### Clec9A binds directly and specifically to RNF41

To identify novel conserved Clec9A interactions, we generated ectodomains of mouse and human Clec9A (mClec9A, hCLEC9A), which encompassed their full extracellular regions (ecto) including the C-type lectin-like domain (CTLD) and stalk (*Zhang et al., 2012*). As mouse and human Clec9A share 69% similarity, we hybridized mClec9A-ecto with protein microarrays consisting of glutathione S-transferase (GST)-tagged human proteins. We found mClec9A-ecto bound to human RNF41, whereas an independent control C-type lectin, Cire/DC-SIGN, did not bind RNF41 (*Supplementary file 1*). In addition, mClec9A-ecto did not bind to 29 other members of the RNF family present on the arrays, indicating a specific interaction between Clec9A and RNF41.

The E3 ubiquitin ligase RNF41 is highly conserved across species, with only one amino acid difference between human and mouse RNF41. To validate the Clec9A-RNF41 interaction, we expressed GST-tagged full-length human RNF41 (<u>R</u>ING, <u>B</u>-box, <u>C</u>oiled-coil, <u>C</u>-terminal domain) in bacterial cells, immobilized it onto glutathione beads, and investigated its ability to bind Clec9A. We found that mClec9A-ecto directly bound to RNF41, but not to control GST protein (*Figure 1—figure supplement 1A,B*). By contrast, independent control C-type lectin ectodomain, Clec12A-ecto (*Zhang et al., 2012*) did not bind to RNF41 (*Figure 1—figure supplement 1A,B*). To determine the RNF41 domains required for interaction with Clec9A, we expressed a series of truncated RNF41 proteins (*Figure 1—figure supplement 1A*). mClec9A-ecto bound the C-terminal domain of RNF41 (RNF41-C), but not the N-terminal domains (RNF41-RBC) (*Figure 1—figure supplement 1B*). This was confirmed using an enzyme-linked immunosorbent assay (ELISA); both mClec9A-ecto and hCLEC9A-ecto bound to RNF41 C-terminal domains, but not to N-terminal domains or GST controls, whereas the control mClec12A-ecto showed no binding to any of the RNF41 constructs (*Figure 1A, B* and *Figure 1—figure supplement 1C,D*). This confirmed that Clec9A binds specifically to the RNF41 C-terminal domain and that the Clec9A-RNF41 interaction is conserved in humans and mice.

### Clec9A-RNF41-binding properties

Clec9A recognition of dead cells via binding to its ligand F-actin, requires conserved tryptophan residues located in the Clec9A-CTLD (hCLEC9A W131 and W227; mClec9A W155 and W250) (*Zhang et al., 2012*). Accordingly, we examined whether Clec9A binding to RNF41 was similarly mediated. Wild-type and mutated hCLEC9A (W131A, W227A) bound comparably to RNF41, indicating that the Clec9A residues required for binding F-actin differed from those required for binding RNF41 (*Figure 1—figure supplement 1C,D*). We therefore explored whether the two ligands compete for binding to Clec9A. RNF41 bound to Clec9A comparably both in the presence or absence of actin (*Figure 1—figure supplement 1E,F*) indicating that actin and RNF41 do not compete for Clec9A binding. To further determine the regions of Clec9A involved in binding RNF41, we performed inhibition studies using the anti-Clec9A antibody 10B4, which is directed against the long loop region of the mClec9A-CTLD. Pre-incubation of Clec9A with 10B4 partially inhibited Clec9A-RNF41 binding, suggesting that Clec9A binding to RNF41 was mediated, at least in part, via the Clec9A CTLD long loop region (*Figure 1—figure supplement 1G,H*).

Clec9A exists as a disulfide-bonded homodimer, which can adopt either a reversible reduction-sensitive (type-1) or a reduction-resistant (type-2) dimer conformation (*Hanč et al., 2016*). At lower pH, both mouse and human Clec9A adopt a reduction-resistant dimer conformation. This reduction-resistant conformation promotes cross-presentation of dead cell-derived Ag, although it does not affect the ability of Clec9A to bind its ligand F-actin, to signal through Syk kinase or to undergo internalization (*Hanč et al., 2016*). Thus, we postulated that the reduction-resistant conformation of

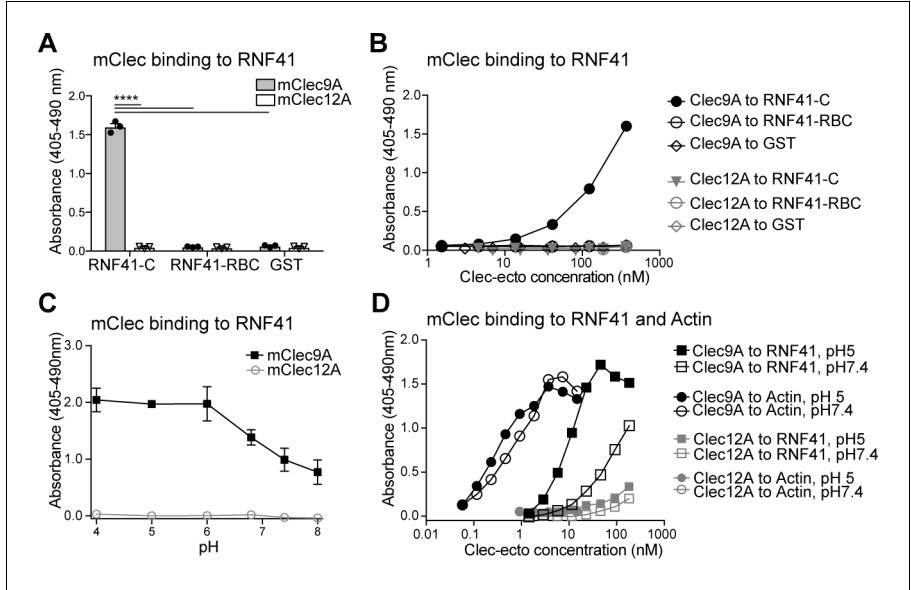

**Figure 1.** Clec9A binds to the E3 Ubiquitin ligase RNF41. (**A, B**) Mouse Clec9A binds to the RNF41 C-terminal domain. ELISA plates were coated with GST-tagged RNF41-C-terminal domain (RNF41-C), RNF41-N-terminal domains (RNF41-RBC) and GST control. (**A**) Binding of FLAG-tagged mClec9A-ecto or the control FLAG-mClec12A-ecto (10 µg/ml) was detected with anti-FLAG HRP. Cumulative data of three experiments is shown, presented as mean ± SEM (unpaired t-test) ****p<0.0001. (**B**) Binding of FLAG-tagged mClec9A-ecto or the control FLAG-mClec12A-ecto (10 µg/ml to 0.04 µg/ml) was detected with anti-FLAG HRP. Cumulative data of three experiments is shown. (**C, D**) Mouse Clec9A binding to RNF41 is enhanced at low pH. ELISA plates were coated with GST-tagged RNF41-C, actin complexes or GST. Binding of biotinylated FLAG-tagged mClec9A-ecto or biotinylated control FLAG-mClec12A-ecto at different pH was detected with streptavidin-HRP. Data is presented as the absorbance 405–490 nm of (Clec binding to GST-RNF41 - Clec binding to GST). (**C**) Cumulative data of three experiments is shown, presented as mean ± SEM. mClec9A (2.5 µg/ml) binding to RNF41-C at pH 4, pH 5 and pH 6 was significantly greater than at pH 7.4 and at pH 8 (1-way ANOVA) *p<0.05. (**D**) Representative of three independent experiments.

The online version of this article includes the following source data and figure supplement(s) for figure 1:

**Source data 1.** Clec9A binding to RNF41.

**Figure supplement 1.** Clec9A binds to the E3 ubiquitin ligase RNF41.

---

Clec9A may promote cross-presentation via its interaction with RNF41. We observed that Clec9A binding to RNF41 was potentiated at lower pH (*Figure 1C,D*), suggesting that the enhanced binding of the reduction-resistant Clec9A dimer, such as found in low pH endosomes, to RNF41, may play a role in cross-presentation.

## Clec9A interacts with RNF41 in cross-presenting DC

Having established that RNF41 can directly interact with the Clec9A ectodomain in vitro, we next explored whether this interaction occurred within the mouse cDC1 that selectively express Clec9A. We validated the specificity of antibodies for RNF41 and Clec9A (*Figure 2—figure supplement 1A–D*) and confirmed RNF41 expression in DC generated by bone marrow Flt3L-culture (Flt3L-DC) and in the cDC1 cell line MutuDC 1940 (*Fuertes Marraco et al., 2012*; *Figure 2A* and *Figure 2—figure supplement 1E*). RNF41 has a putative-N-terminal myristoylation site, thought to anchor RNF41 to intracellular membranes (*Masschaele et al., 2017*; *Masschaele et al., 2018*). Cellular fractionation studies of cDC1 indicated the major RNF41 isoform (~39 kDa) was primarily associated with digitonin soluble fractions containing cytosol and early endosomes (indicated by GAPDH and EEA-1 expression), but was also associated with detergent (IGEPAL)-resistant membrane fractions (*Figure 2A*). We also observed a larger RNF41 isoform (~55 kDa) that was differentially associated with IGEPAL soluble membrane fractions, including endoplasmic reticulum (ER) and recycling endosomes (indicated by Calreticulin and Rab11 expression). Consistent with its role as an endocytic membrane

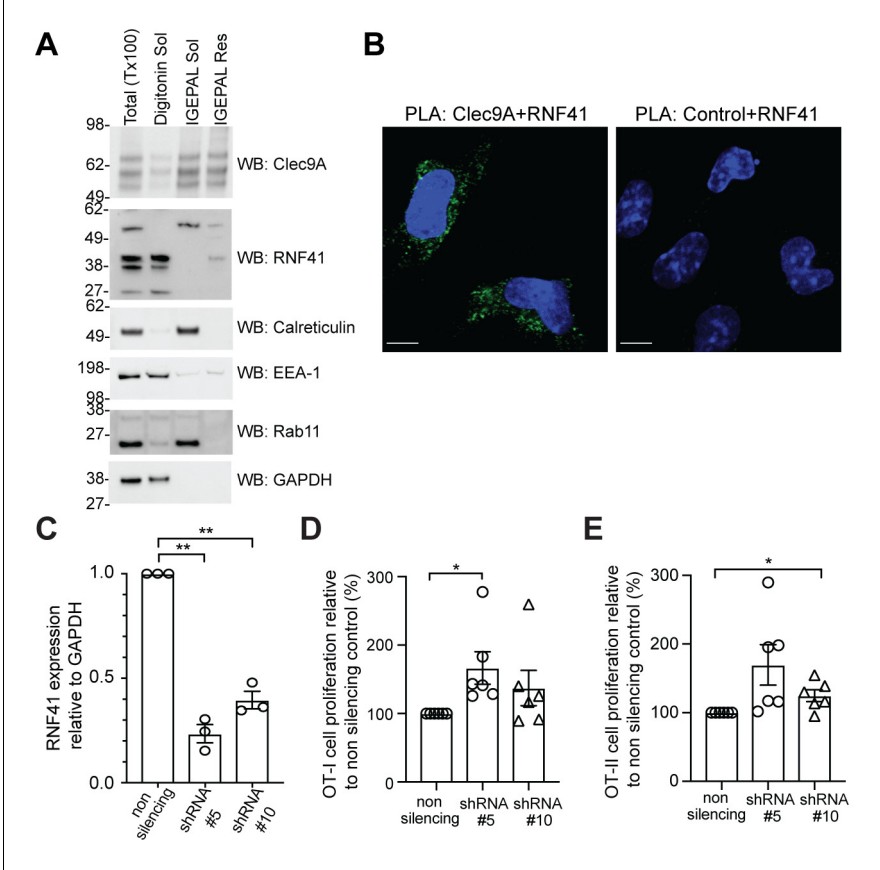

**Figure 2.** RNF41 regulates presentation of dead cell-associated Ag in DC. (**A**) RNF41 is associated with membrane-associated cellular fractions. Cellular fractions were prepared from MutuDC 1940 by sequential detergent lysis and centrifugation conditions. In brief, DC were initially treated with digitonin and centrifugation, and digitonin soluble fractions harvested (containing GAPDH, EEA-1). Digitonin resistant fractions were then subjected to IGEPAL lysis and centrifugation, and IGEPAL soluble fractions (containing Calreticulin and Rab11) and IGEPAL resistant fractions harvested. Total lysates (Tx100) or cellular fractions (Digitonin soluble, Sol; IGEPAL soluble, Sol; IGEPAL resistant, Resistant) were analyzed by western blot (WB). Representative of five independent experiments. (**B**) Clec9A and RNF41 interact in DC. A proximity ligation assay (PLA) was performed using MutuDC 1940 with mouse polyclonal anti-Clec9A serum directed against the extracellular domain, or control unimmunized serum, and rabbit anti-RNF41 polyclonal Ab. PLA signal is shown in green, DAPI (nucleus) in blue. Images are representative of four independent experiments (Clec9A+RNF41: 214 cells, Control+RNF41: 168 cells). Scale bars represent 10 μm. (**C–E**) RNF41 knockdown enhances presentation of dead cell Ag. MutuDC 1940 were transduced with shRNA construct #5 or #10 to knockdown RNF41 expression, compared with a non-silencing control retrovirus. (**C**) Reduced RNF41 expression was confirmed by qPCR. Cumulative data from three independent experiments, demonstrating the mean ± SEM (paired t-test) **p<0.01. (**D–E**) Transduced MutuDC 1940 were cultured with dead CHO-K1 cells expressing OVA at a ratio of 1 DC: 2 dead CHO-K1 cells, followed by culture with (**D**) OT-I and (**E**) OT-II Transgenic T cells for 3 days. T cell proliferation relative to the non-silencing control. Cumulative data from six independent experiments, demonstrating the mean ± SEM (paired t-test) *p<0.05. The online version of this article includes the following source data and figure supplement(s) for figure 2:

**Source data 1.** RNF41 knockdown: RNF41 expression and Ag presentation.
**Figure supplement 1.** Detection of RNF41, Clec9A and their interaction.
**Figure supplement 2.** Subcellular localization of RNF41 in DC by immuno-electron microscopy.
**Figure supplement 3.** RNF41 regulates Clec9A and the presentation of dead cell-associated Ag in DC.

receptor, Clec9A was found in Digitonin soluble fractions containing early endosomes, in IGEPAL soluble fractions containing recycling endosomes and in detergent-resistant membrane fractions. Microscopy studies of cDC1 confirmed RNF41 expression both in the cytoplasm and associated with endosomal vesicles, as indicated by colocalization with EEA-1 and Rab11, respectively (*Figure 2—*

*figure supplement 1E*). We further demonstrated localization of RNF41 to the cytoplasm, the plasma membrane and intracellular membranes such as ER, early endosomes, late endosomes, endosomal tubular-vesicular compartments, based on the morphological characteristics of these compartments using immuno-electron microscopy (*Figure 2—figure supplement 2*). Thus, our data indicate that Clec9A and RNF41 can be found in similar types of membrane-associated intracellular compartments.

We next examined whether Clec9A and RNF41 interact in cDC1 using a Proximity Ligation Assay (PLA). RNF41 and the extracellular domain of Clec9A colocalized within 40 nm in cDC1, indicative of an in situ Clec9A-RNF41 interaction within DC (*Figure 2B*). Treatment with the proteasome inhibitor MG132 enhanced Clec9A and RNF41 interactions within steady-state cDC1 (*Figure 2—figure supplement 3A*: *No dead cells*), suggesting that one outcome of the Clec9A-RNF41 interaction may be for RNF41 to ubiquitinate and target Clec9A, or its associated Ag cargo, for proteasomal degradation.

## RNF41 regulates antigen presentation of dead cell-associated antigens

Clec9A facilitates processing of Ag for presentation on both MHC I and MHC II (*Caminschi et al., 2008*; *Lahoud et al., 2011*; *Sancho et al., 2008*; *Tullett et al., 2016*; *Zelenay et al., 2012*), prompting us to investigate whether RNF41 regulates processing and presentation of dead cell-associated Ag. To examine the dose dependence of RNF41, we utilized two shRNA constructs (shRNA #5, shRNA #10) that produced knockdown of RNF41 at the RNA level to differing degrees, compared to the non-silencing control (*Figure 2C*). Knockdown of RNF41 in cDC1 using shRNA#5 resulted in significantly enhanced cross-presentation of dead cell-associated Ag (*Figure 2D* and *Figure 2—figure supplement 3B*), whereas the modest reduction of RNF41 using shRNA #10 only resulted in a slight enhancement of cross-presentation that did not reach statistical significance (*Figure 2D*). Similarly, knockdown of RNF41 resulted in enhanced MHC II presentation of dead cell-derived Ag (*Figure 2E* and *Figure 2—figure supplement 3C*), albeit the absolute level of CD4 T cell stimulation by cDC1 was lower than that of CD8 T cells, as expected for cDC1 (*Rizzitelli et al., 2006*). This data suggested that RNF41 down-regulates the processing of dead cell-associated Ag for both MHC II presentation and MHC I cross-presentation.

## RNF41 regulates degradation of the Clec9A receptor

We investigated whether RNF41 mediated its effects on Ag presentation via regulation of the steady-state levels of the Clec9A receptor, and targeting Clec9A for degradation, as previously described for other RNF41-receptor associations (*Qiu and Goldberg, 2002*). RNF41 and FLAG-tagged Clec9A were co-expressed in 293 F cells, which do not express endogenous Clec9A but do express low levels of endogenous RNF41. We confirmed RNF41 and Clec9A can interact in transfected 293 F cells by co-immunoprecipitation (IP) studies (*Figure 2—figure supplement 1D*). Co-transfection of RNF41 and Clec9A resulted in reduced Clec9A levels, both at the level of total cellular expression and at the cell surface (*Figure 3A,B*). By contrast, co-expression of a control E3 ubiquitin ligase RNF6 did not affect Clec9A levels (*Figure 3A,B*). RNF41 regulation of both mouse and human Clec9A levels was dependent on the RING domain that is necessary for ubiquitin transfer, as expression of full-length RNF41 resulted in reduced Clec9A levels, whereas the truncated RNF41 (RNF41-ΔRING) that lacks the RING domain, had no effects on Clec9A levels (*Figure 3C,D*). RNF41 did not regulate levels of control receptor Clec12A, confirming specificity of RNF41 regulation of Clec9A (*Figure 3C*). Overall, this suggests that RNF41-mediated ubiquitination of Clec9A promotes its degradation, most likely at the ER or endosomal compartments, and that this restrains Ag presentation.

As the interaction between Clec9A-RNF41 was mediated via the Clec9A extracellular domain, we next investigated whether the transmembrane and intracellular domains of Clec9A were necessary for RNF41 regulation of Clec9A. Co-transfection of RNF41 with a recombinant form of Clec9A, which includes extracellular domains but lacks transmembrane and intracellular domains (Clec9A-ecto), resulted in reduced Clec9A-ecto levels (*Figure 3—figure supplement 1A*). This suggested that RNF41 interaction with the extracellular domains of Clec9A is sufficient to regulate Clec9A fate. We further excluded the possibility that Clec9A might interact with RNF41 released from damaged or

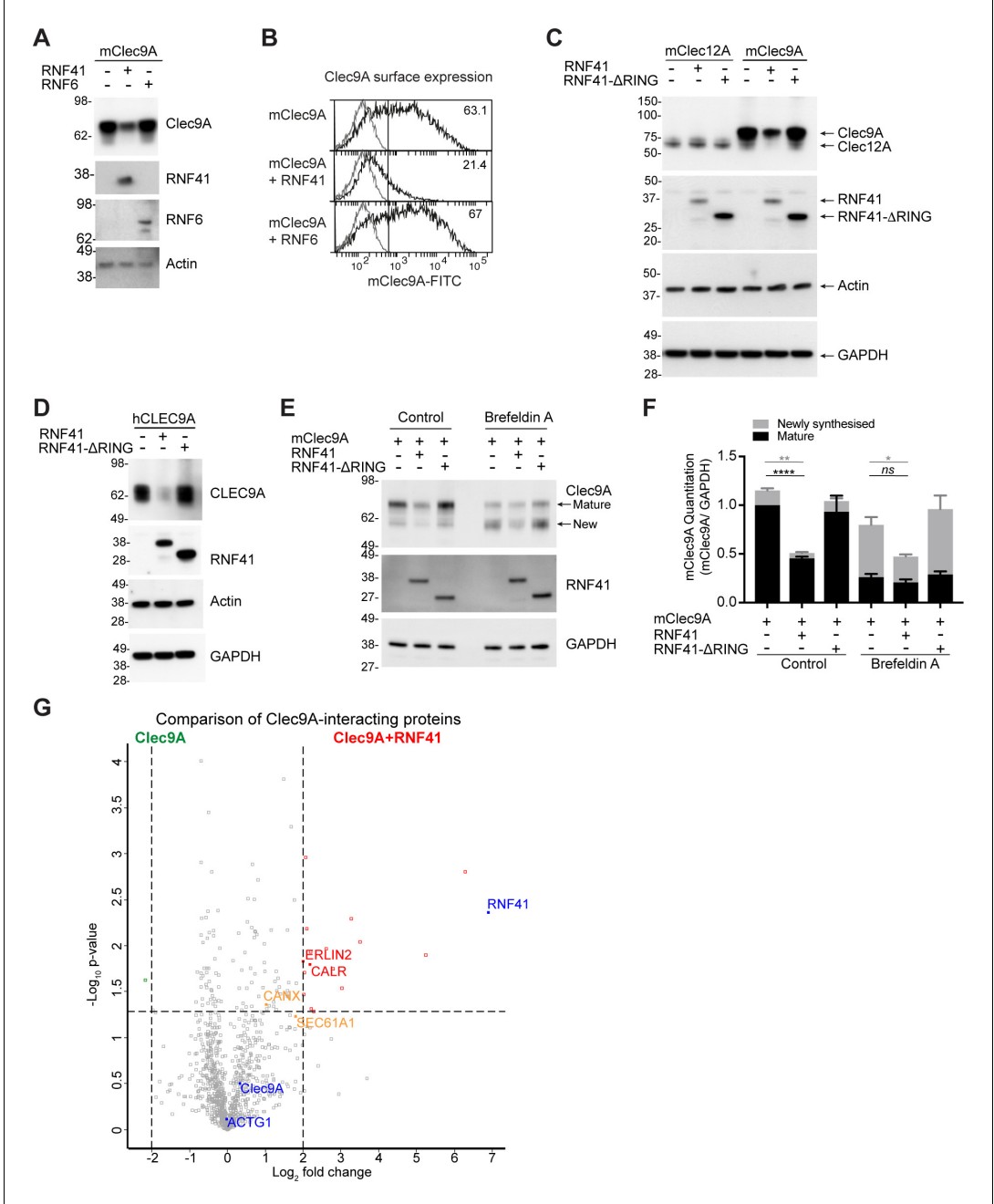

**Figure 3.** RNF41-mediated regulation of Clec9A receptor levels is associated with novel interactions. (**A, B**) RNF41 co-expression reduces Clec9A expression. 293 F cells were co-transfected with FLAG-tagged full-length mClec9A (mClec9A-FLAG), RNF41 or RNF6-Myc. (**A**) Cellular lysates were analyzed by WB (24 hr). (**B**) Clec9A surface expression of transfected cells was analyzed by flow cytometry (Transfected Clec9A: solid line; untransfected: gray line; 48 hr). Representative of three independent experiments. Cells transfected with Clec9A and RNF41 had significantly reduced Clec9A surface levels compared with cells transfected with Clec9A; cumulative data from six experiments (paired t-test) **p<0.01. (**C, D**) RNF41 regulation of Clec9A is mediated by the RNF41 RING domain. 293 F cells were co-transfected with mClec9A-FLAG/mClec12A-FLAG/hCLEC9A-FLAG, and RNF41 or RNF41-Δ RING. Cellular lysates were analyzed by WB. Representative of (**C**) 8 and (**D**) three independent experiments. (**E, F**) RNF41 regulates newly synthesized and mature Clec9A. 293 F cells were co-transfected with mClec9A-FLAG and RNF41 or RNF41-ΔRING. At 14 hr post-transfection, cells were treated with Brefeldin A (5 µg/ml) or DMSO for a further 4 hr. (**E**) Cellular lysates were analyzed by WB. (**F**) Densitometric analysis of newly synthesized and mature Clec9A levels relative to GAPDH from four experiments is shown, demonstrating Clec9A levels as mean ± SEM (paired t-test). *ns* not significant, *p<0.05, **p<0.01, ****p<0.0001. (**G**) Identification of novel Clec9A-interaction partners that are enhanced by RNF41. 293 F cells were co-transfected with mClec9A-FLAG in the presence or absence of RNF41. At 22 hr post-transfection Clec9A-interacting proteins were immunoprecipitated (IP) and analyzed by LC-MS/MS. Volcano plot (X-axis: log2 fold-change; Y-axis: -Log10 p-value) comparing Clec9A-interacting proteins from cells transfected

*Figure 3 continued on next page*

Figure 3 continued

with mClec9A+RNF41 versus mClec9A alone. The dotted vertical lines indicate a 4-fold protein change. The dotted horizontal line indicates a p-value cut-off of 0.05. Selected proteins with >4 fold upregulation are annotated in red,>2 fold upregulation in orange,>4 fold downregulation in green. Clec9A and its interacting proteins, actin (ACTG1) and RNF41, are in blue.

The online version of this article includes the following source data and figure supplement(s) for figure 3:

**Source data 1.** RNF41 regulation of Clec9A.
**Figure supplement 1.** RNF41 mediated regulation of Clec9A is associated with novel interactions.

neighboring cells, indicating that it is the co-expression of Clec9A and RNF41 within the same cell that facilitates RNF41 regulation of Clec9A (*Figure 3—figure supplement 1B,C*).

The role of RNF41 in regulating newly synthesized Clec9A was investigated by monitoring Clec9A protein maturation via glycan modification. mClec9A can be modified by N-linked glycosylation at N81 in the stalk region and N159 in the CTLD (*Caminschi et al., 2008*). Using western blot analysis under non-reducing conditions and Liquid Chromatography Tandem Mass-Spectrometry (LC-MS/MS), we identified two forms of the Clec9A dimer: a ~ 75 kDa mature protein with post ER-glycosylation, and a ~ 60 kDa form with less complex, high mannose, glycosylation indicative of a newly synthesized protein (*Figure 3E*, *Supplementary file 1* and *Figure 3—figure supplement 1D*). To determine whether RNF41 acts on Clec9A at the ER, we used Brefeldin A to inhibit transport of newly synthesized proteins from the ER/Golgi to the cell surface. This resulted in a reduction of the mature Clec9A dimer, and an increase in the newly synthesized Clec9A dimer (*Figure 3E* and *Figure 3—figure supplement 1E*). However, Brefeldin A did not inhibit RNF41-mediated degradation of Clec9A (*Figure 3E,F*). Moreover, co-expression of Clec9A with RNF41, in the presence of Brefeldin A, resulted in reduced levels of the newly synthesized ER-resident form of Clec9A (*Figure 3E,F*). These data demonstrate that as well as potentially interacting in endosomal compartments (*Figure 2—figure supplements 1E* and *2*), RNF41 can also regulate levels of newly synthesized Clec9A at the ER.

## Characterization of the Clec9A-RNF41 interactome

To elucidate the molecular machinery that facilitates access of Clec9A ectodomains to degradation machinery, we performed a proteomic analysis of Clec9A-interacting proteins in transfected cells. FLAG-tagged mClec9A was immunoprecipitated from cells transfected with Clec9A in the presence or absence of RNF41, digested with trypsin and subjected to LC-MS/MS analysis. As expected, most Clec9A-interacting proteins were identified at comparable levels in the presence or absence of RNF41. However, in the presence of full-length RNF41, we observed a 4-fold or greater increase in the abundance of 17 Clec9A-interacting proteins, considering a false discovery rate (FDR) cutoff of 0.05 (*Figure 3G*, *Figure 3—figure supplement 1F* and *Supplementary file 1*). Focusing on ER-associated proteins, we identified an increased association of Clec9A with ER membrane protein Erlin2 (4-fold), ER chaperones calreticulin (CALR >4 fold) and calnexin (CANX >2 fold), and with translocon Sec61A1 (>3.5 fold), supporting a role for RNF41 in ER-associated regulation of Clec9A in steady state. Comparison of Clec9A-interacting proteins in the presence of RNF41 versus RNF41-Δ RING indicated that most of these Clec9A-interactions, including Erlin2 and Calreticulin, may be enhanced by ubiquitination mediated by the RING domain, rather than RNF41 binding per se (*Figure 3—figure supplement 1G* and *Supplementary file 1*). It further revealed an increased Clec9A interaction with proteasomal complex component PSMB1, confirming RNF41-mediated regulation of Clec9A can be associated with proteasomal degradation. An analysis of these Clec9A-interacting molecules identified novel pathways and potential candidates for chaperoning Clec9A in the ER and for proteasomal degradation (*Figure 3—figure supplement 1H*).

We initially focused on the ER-associated proteins Erlin2 and Erlin1 (*Browman et al., 2006*), identified as potentially interacting with Clec9A (*Figure 3—figure supplement 1F,G*), as they are highly expressed by cDC1 (*Theisen et al., 2018*) and have been shown to facilitate recruitment of E3 ubiquitin ligases to receptors (*Lu et al., 2011*). We validated the novel Clec9A interaction with endogenous Erlin1 and Erlin2 by co-IP studies of Clec9A transfected 293 F cells (*Figure 3—figure supplement 1I,J*). We further demonstrated that the Clec9A-Erlin1/2 interactions were increased in the presence of RNF41 in a RING domain-dependent manner (*Figure 3—figure supplement 1I,J*),

although the total levels of endogenous Erlin1 and Erlin2 did not change (*Figure 3—figure supplement 1I*: Input samples). Given that the RNF41 RING domain is required for RNF41-mediated regulation of Clec9A, these data favor a model in which the Clec9A-Erlin 1/2 interaction is associated with RNF41 regulation of Clec9A.

## RNF41 regulates ubiquitination of Clec9A

RNF41 can directly ubiquitinate target proteins, marking them for downstream fates including proteasomal degradation and trafficking. To test whether RNF41 regulates Clec9A via ubiquitination, we analyzed transfected cells co-expressing RNF41 and Clec9A. As expected, RNF41 mediated poly-ubiquitination of mClec9A and hCLEC9A, whereas RNF41-ΔRING did not mediate increased ubiquitination of Clec9A above the levels observed by transfection with Clec9A alone (*Figure 4A,B*). Similarly, purified RNF41 was able to mediate ubiquitination of Clec9A in vitro (*Figure 4C*).

To confirm and extend these findings, we utilized LC-MS/MS to identify the amino acid residues within Clec9A ubiquitinated by RNF41. FLAG-tagged mClec9A was immunoprecipitated from cells co-expressing either full-length RNF41 or RNF41-ΔRING (which is unable to mediate Clec9A ubiquitination), digested with trypsin and subjected to LC-MS/MS analysis. Ubiquitination sites on Clec9A were identified by the presence of the characteristic di-glycine footprint motif (Gly-Gly; +114.03 Da mass shift), which is left behind on ubiquitinated lysine residues after trypsin digestion. This analysis revealed several ubiquitination sites within the ectodomain of mClec9A including K88 in the stalk region, and K167, K178, K195, K238, K246, K251, K257 and K258 within the CTLD (*Figure 4D* and *Figure 4—figure supplement 1*), most of which are conserved in mouse and human Clec9A. In addition, we observed an increase in abundance of ubiquitinated Clec9A peptides in the presence of RNF41, but not RNF41-ΔRING (*Figure 4E*, and *Supplementary file 1*). These results confirm that RNF41 ubiquitinates the Clec9A ectodomain, and that the RNF41 RING domain is essential for ubiquitination.

The nature of the ubiquitin linkage-specific modification influences the fate of the ubiquitinated target protein: K-48 ubiquitin linkages are generally associated with targeting proteins for proteasomal degradation, whereas K-63 linkages have been reported to facilitate non-proteasomal functions including endocytic trafficking. Using a K-48 linkage-specific ubiquitin antibody, we detected K-48 linkages on ubiquitin associated with Clec9A suggesting that Clec9A poly-ubiquitination could be mediated, at least in part, via K48-linkages (*Figure 4A*). LC-MS/MS analysis of di-glycine motifs on ubiquitin molecules associated with Clec9A confirmed this notion and also revealed an increase in abundance of K48-linked ubiquitin in a RNF41-dependent manner (*Figure 4E* and *Supplementary file 1*). In addition, we detected increased Clec9A-associated K11- and K63-linked ubiquitination (*Figure 4E* and *Supplementary file 1*), which suggests RNF41 poly-ubiquitination of Clec9A may include all three ubiquitin linkages or mixed ubiquitin chains.

Taken together, our data demonstrates that RNF41 regulates ubiquitination of Clec9A in steady-state DC to restrain Clec9A and thereby presentation of dead cell-associated Ag. These data indicate that RNF41 may itself be a checkpoint that ensures Ag cross-presentation is controlled in time and space.

## Dead cell uptake diverts Clec9A from a degradative pathway

To further investigate the mechanism by which RNF41 could mediate down-regulation of dead cell-associated Ag presentation in DC, we investigated the interaction of Clec9A with RNF41 following dead cell uptake. cDC1 were cultured with PKH26-labeled CHO-K1 dead cells; Clec9A was associated with dead cell material from 2 to 5 hr, consistent with its role in intracellular trafficking of internalized dead cell material (*Zelenay et al., 2012*; *Figure 5—figure supplement 1A,B*). To determine the extent of the intracellular Clec9A-RNF41 interaction during DC processing of dead cells, we utilized PLA. The degree of Clec9A-RNF41 interaction was significantly reduced in the presence of dead cells at 4 hr both in the cDC1 cell line MutuDC 1940 and in primary splenic cDC1 (*Figure 5A* and *Figure 5—figure supplement 1C*), although neither Clec9A nor RNF41 levels were reduced (*Figure 5—figure supplement 1D*). Addition of the proteasome inhibitor MG132 did not rescue Clec9A-RNF41 PLA signals in the presence of dead cells (*Figure 2—figure supplement 3A*), suggesting that the reduced Clec9A-RNF41 interaction early after dead cell uptake, reflects molecular

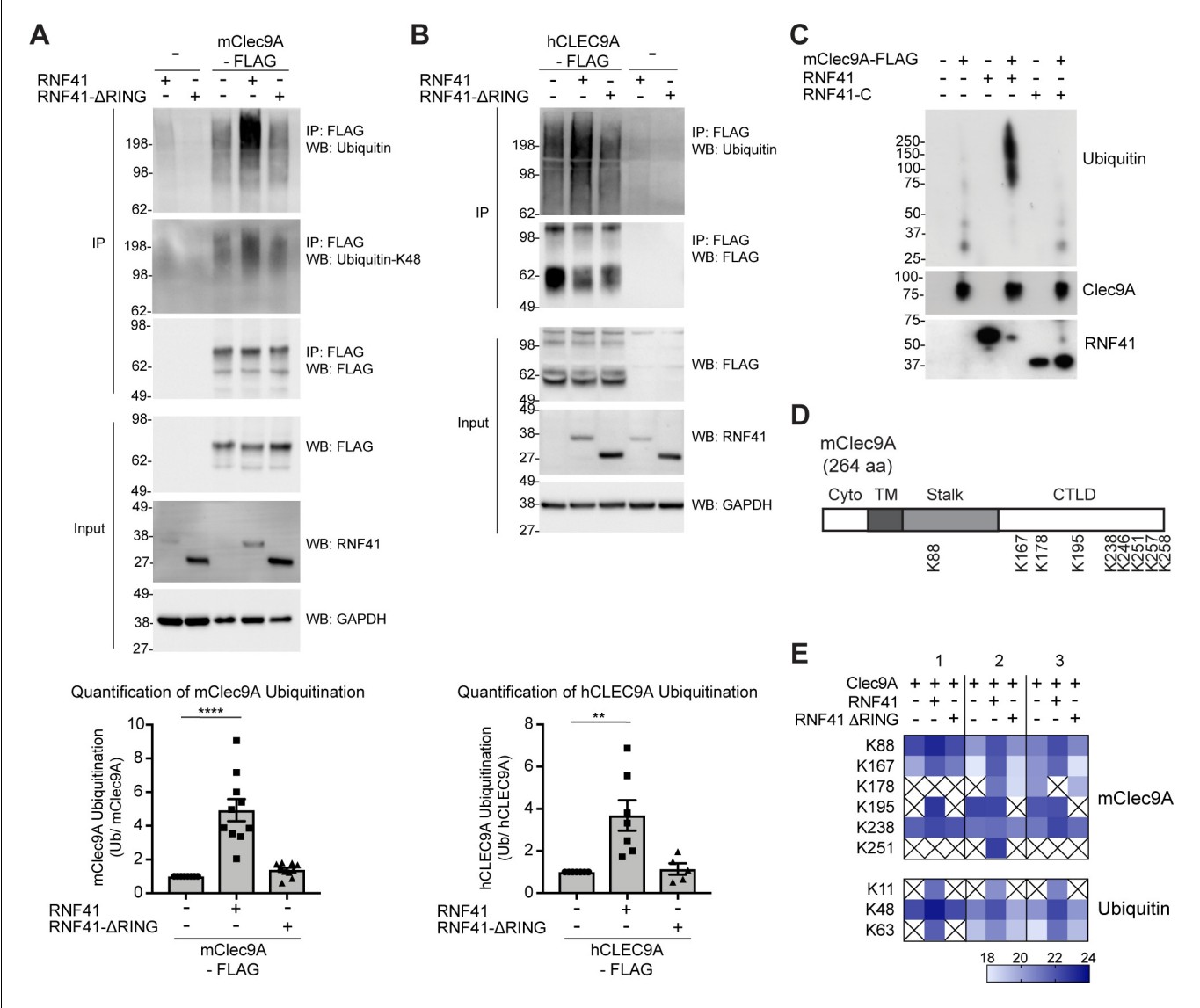

**Figure 4.** RNF41 interacts with Clec9A to mediate Clec9A ubiquitination. (**A, B**) RNF41 mediates mClec9A ubiquitination in vivo. 293 F cells were co-transfected with mClec9A-FLAG or hCLEC9A-FLAG, Myc-tagged ubiquitin (Ub-Myc), and RNF41 or RNF41-ΔRING. At 22 hr, IP Clec9A complexes were analyzed for ubiquitination by WB using anti-Ubiquitin Ab, representative of (**A**) 10 experiments, (**B**) seven experiments, and anti-Ubiquitin K-48 Ab, representative of (**A**) two experiments. Densitometric analysis of ubiquitinated Clec9A relative to total Clec9A from IP of Clec9A complexes from (**A**) 10 experiments and (**B**) seven experiments is shown, as mean ± SEM (ANOVA) **p<0.01, ****p<0.0001. (**C**) RNF41 mediates Clec9A ubiquitination in vitro. mClec9A-FLAG was immunoprecipitated from transfected 293 F cells, then incubated with GST-RNF41 or GST-RNF41-C in the presence of E1, E2 (UbcH5a) and biotinylated ubiquitin. Ubiquitination of Clec9A complexes was analyzed by WB for ubiquitin (Streptavidin-HRP), total Clec9A (anti-FLAG-M2) and total RNF41. Representative of two independent experiments. (**D, E**) Proteomic analysis of RNF41 mediated ubiquitination of Clec9A complexes. 293 F cells were co-transfected with mClec9A-FLAG, Ub-Myc and RNF41 or RNF41-ΔRING. At 22 hr, IP Clec9A complexes (pH 6.0) were analyzed by LC-MS/MS. (**D**) Nine major Clec9A ubiquitination sites were identified, from at least four independent experiments, and the position of these sites is indicated. (**E**) Heat map showing quantification results of 6 and 3 ubiquitinated peptides derived from Clec9A and Ubiquitin, respectively, across three independent experiments as indicated by 1,2,3 above the heat map. The intensities of the ubiquitinated peptides are shown as log2. An X indicates that the ubiquitinated peptide was not confidently detected or quantified.

The online version of this article includes the following source data and figure supplement(s) for figure 4:

**Source data 1.** RNF41 mediates ubiquitination of mouse and human Clec9A.

**Figure supplement 1.** Identification of ubiquitination sites for Clec9A.

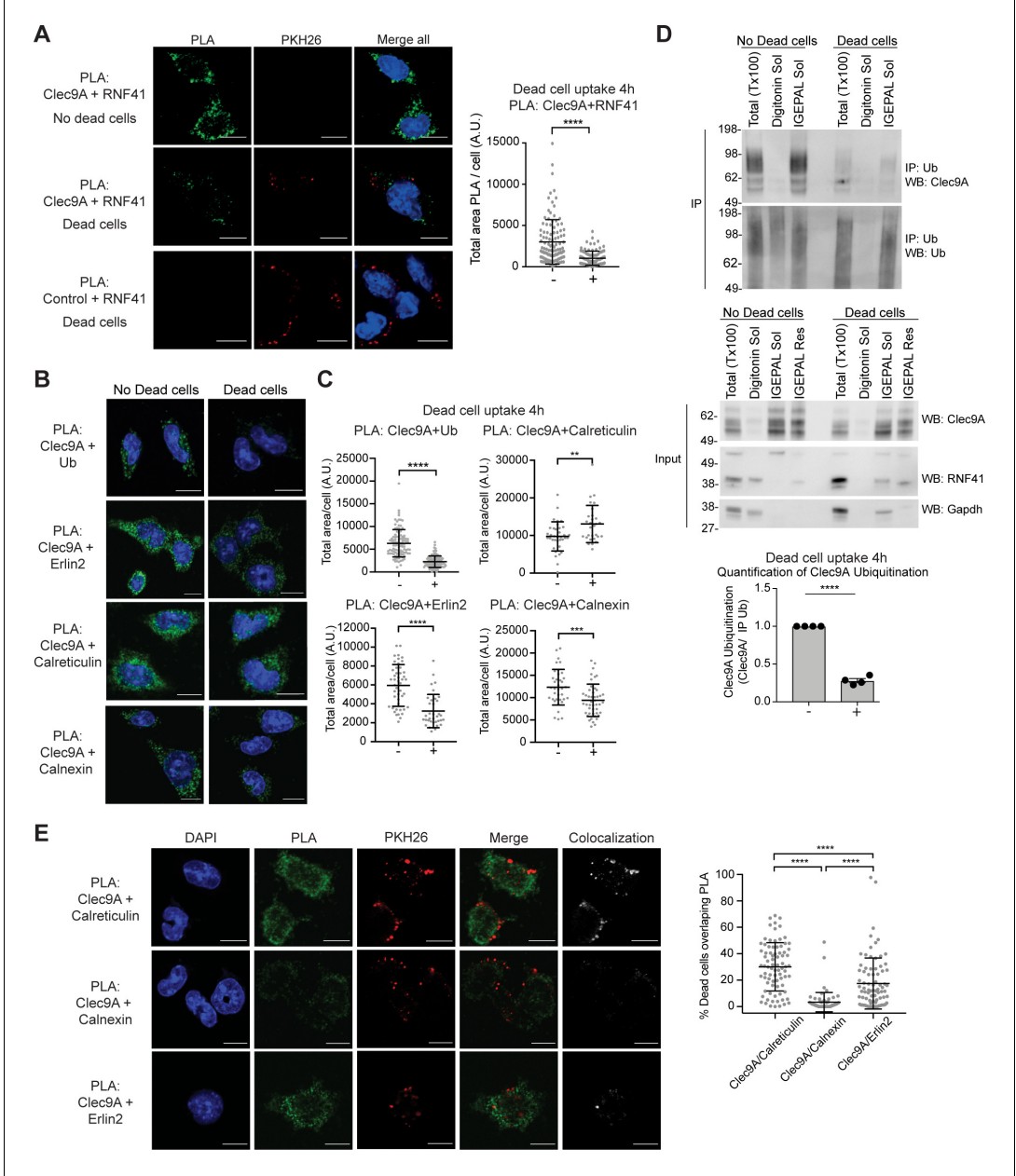

**Figure 5.** Dead cell uptake diverts Clec9A away from a degradative pathway. (**A**) Clec9A and RNF41 interaction in DC is reduced at early stages following dead cell uptake. MutuDC 1940 were cultured in the absence (-) and presence (+) of PKH26-labeled CHO-K1 dead cells for 4 hr, and a PLA performed for Clec9A and RNF41. Representative images from three independent experiments. PLA signal is in green, dead cells (PKH26) in red, DAPI in blue. Scale bars represent 10 μm. The total area of Clec9A+RNF41 PLA per cell (Arbitrary Units, A.U.) was analyzed by Mann–Whitney test. Cumulative data from three independent experiments (4 h-dead cells: 112 cells; 4 h+dead cells: 96 cells). Bars represent mean ± SD. ****p<0.0001. (**B–C**) Clec9A-Ubiquitination and Clec9A interactions in DC are differentially regulated in the presence of dead cells. MutuDC 1940 were cultured +/− CHO-K1 dead cells for 4 hr, and a PLA performed for Clec9A with Ubiquitin, Erlin2, Calreticulin or Calnexin. (**B**) Representative images from two independent experiments (Clec9A+Erlin2 PLA: no dead cells; 85, +dead cells: 83 cells. Clec9A+Calreticulin PLA; no dead cells: 83 cells, +dead cells: 81 cells. Clec9A+Calnexin PLA; no dead cells: 84 cells, +dead cells: 73 cells; Clec9A+Ubiquitin PLA: no dead cells;101, + dead cells; 106 cells). PLA is in green, DAPI in blue. Scale bars represent 10 μm. (**C**) The total area of PLA per cell (A.U.) was analyzed by Mann–Whitney test (Clec9A+Ub PLA; no dead cells: 101 cells; +dead cells: 106 cells. Clec9A+Erlin2 PLA; no dead cells: 46 cells, +dead cells: 36 cells. Clec9A+Calreticulin PLA; no dead cells: 36 cells, +dead cells: 29 cells.) and unpaired t-test (Clec9A+Calnexin; no dead cells: 37 cells, +dead cells: 46 cells). Data is cumulative from two experiments for Clec9A+Ub; representative of three experiments for Clec9A+Erlin2; representative of two experiments for Clec9A+Calreticulin and Clec9A+Calnexin, Bars represent mean ± SD. **p<0.01, ***p<0.001, ****p<0.0001. (**D**) Clec9A ubiquitination in DC is reduced following dead cell uptake. MutuDC 1940 were cultured +/− CHO-K1 dead cells for 4 hr and cellular fractionation performed. Ubiquitinated proteins were IP using TUBE beads and

*Figure 5 continued on next page*

Figure 5 continued

ubiquitinated Clec9A detected by WB using anti-Clec9A. Representative of four independent experiments. Densitometric analysis of ubiquitinated Clec9A relative to ubiquitinated proteins from IP of ubiquitinated proteins from four experiments is shown, as mean ± SEM (unpaired t-test) ****p<0.0001. (E) Differential association of Clec9A-Calreticulin, Clec9A-Calnexin and Clec9A-Erlin2 interactions with dead cells. MutuDC 1940 were cultured for 4 hr with PKH26-labeled CHO-K1 dead cells, and a PLA performed for Clec9A with Calreticulin, Calnexin and Erlin2. PLA is in green, PKH26-labeled dead cells in red, DAPI in blue. Colocalization images were generated using Image J; pixels that have a positive signal in both PLA and PKH26 are shown in white. Scale bars represent 10 μm. Images are representative of two experiments (PLA Clec9A+Calreticulin: 81 cells; PLA Clec9A +Calnexin: 73 cells; PLA Clec9A+Erlin2: 83 cells). Colocalization of dead cells with PLA was assessed by Manders' coefficient, bars represent mean ± SD (Kruskal-Wallis test with Dunn's multiple comparisons). Cumulative data from two independent experiments ****p<0.0001.

The online version of this article includes the following source data and figure supplement(s) for figure 5:

**Source data 1.** Clec9A interactions are regulated in the presence of dead cells.
**Figure supplement 1.** Dead cell processing alters the Clec9A regulatory pathway.

trafficking rather than proteasomal degradation. By contrast, Clec9A-RNF41 interactions were not reduced at later stages after dead cell uptake (18 hr) (*Figure 5—figure supplement 1E*).

Clec9A ubiquitination, as detected by PLA with antibodies to the Clec9A extracellular domain and to ubiquitin (Ub), was decreased 4 hr following dead cell uptake in cDC1 (*Figure 5B,C*), but appeared comparable in the presence and absence of dead cells by 18 hr (*Figure 5—figure supplement 1F*), consistent with the Clec9A-RNF41 interactions. Biochemical analysis of steady- state cDC1 revealed ubiquitinated Clec9A was primarily associated with IGEPAL soluble membrane fractions, which contain ER (Calreticulin$^+$) and recycling endosomes (Rab11$^+$), but was not detected in digitonin-soluble fractions containing cytosol (GAPDH$^+$) and early endosomes (EEA-1$^+$) (*Figure 5D* and *Figure 5—figure supplement 1G,H*). Furthermore, we observed a significant decrease in ubiquitinated Clec9A in IGEPAL soluble membrane fractions at 2–4 hr following dead cell uptake (*Figure 5D* and *Figure 5—figure supplement 1G,H*), consistent with the reduced Clec9A-RNF41 interaction and the reduced Clec9A ubiquitination observed by PLA at early stages following dead cell uptake (*Figure 5A–C*). Moreover, Clec9A-RNF41 interactions did not appear to be associated with dead cell material at early stages following dead cell uptake (*Figure 5A* and *Figure 5—figure supplement 1I*), suggesting the Clec9A interaction with RNF41 does not play a role in early Clec9A-dependent dead cell processing.

## Clec9A interactions are differentially regulated following dead cell uptake

As Clec9A-Erlin1/2 interactions were associated with RNF41 mediated regulation (*Figure 3G* and *Figure 3—figure supplement 1I,J*), we investigated whether these interactions were regulated in a similar manner to Clec9A-RNF41 interactions and Clec9A ubiquitination in cDC1. We demonstrated Clec9A interaction with Erlin2, by PLA, in steady-state cDC1, validating the biological relevance of this interaction in DC (*Figure 5B*). Following dead cell uptake, we observed a decrease in total levels of Clec9A-Erlin2 interaction, which were not rescued by the proteasomal inhibitor MG132 (*Figure 5B,C*, and *Figure 5—figure supplement 1J*). Furthermore, most Clec9A-Erlin2 PLA signals did not appear associated with dead cell material taken up by the DC (*Figure 5E*). Thus, our data indicates that in steady-state DC, Clec9A and Erlin2 interact, potentially within the ER. However, following dead cell uptake, Clec9A-Erlin2 interactions are reduced, which likely reflects altered Clec9A trafficking events and reduced RNF41-mediated regulation (*Figure 6*).

We next investigated Clec9A-interactions with the ER chaperones Calreticulin and Calnexin as they play a role in MHC-I processing and cross-presentation. We confirmed that Clec9A colocalizes with both Calreticulin and Calnexin in steady-state cDC1 by PLA, suggesting a role for Calreticulin and Calnexin in interacting with, and potentially chaperoning, Clec9A (*Figure 5B,C*). As both Calreticulin and Calnexin can be recruited to Ag-containing endosomes (*Croce et al., 2017*; *Guermonprez et al., 2003*), we examined Clec9A-Calreticulin and Clec9A-Calnexin interactions following dead cell uptake. We observed an increase in Clec9A-Calreticulin interactions by PLA following dead cell uptake (*Figure 5B,C*). Furthermore, the Clec9A-Calreticulin PLA signals appeared to be associated with dead cell material taken up by the DC (*Figure 5E*). By contrast, Clec9A-Calnexin PLA signals did not associate with internalized dead cells within DC (*Figure 5E*). Together, this data suggests a role for both Calreticulin and Calnexin in chaperoning Clec9A in the ER, but suggests

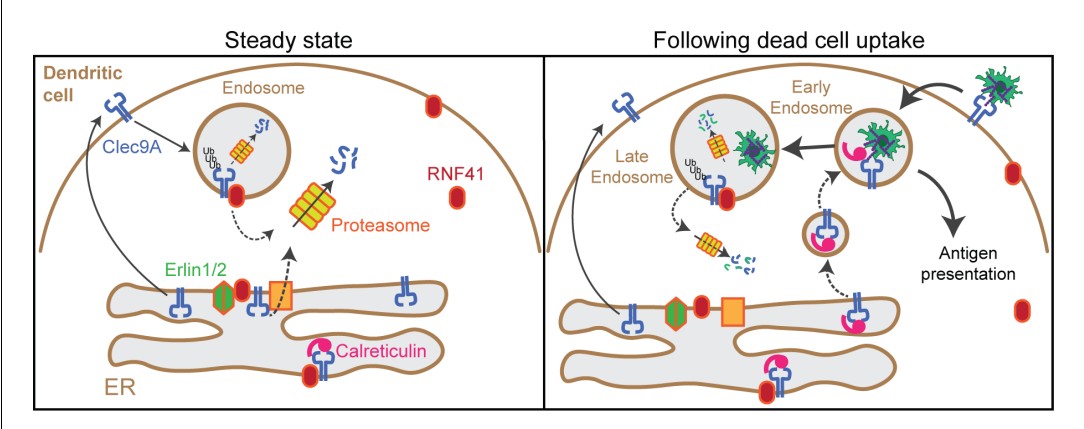

**Figure 6.** Model of RNF41 as a negative regulator of Clec9A and cross-presentation. We propose that at steady state, Clec9A interacts with RNF41, Erlin1/2 and Calreticulin. RNF41 mediates Clec9A ubiquitination at the ER and endosomes, targeting it for degradation to control Clec9A receptor levels. At early stages following dead cell uptake by DC, Clec9A is diverted away from Erlin1/2 and RNF41-mediated degradation, and is recruited with Calreticulin to dead cell-containing endosomes, to facilitate cross-presentation of dead cell Ag. At later stages following dead cell uptake, RNF41 interacts with Clec9A at endosomes to regulate Clec9A following presentation of dead cell Ag.

that it is primarily Calreticulin that chaperones Clec9A or its dead cell cargo to endosomal compartments (*Figure 6*).

Altogether our studies have revealed that RNF41 is a novel regulator of Clec9A and thereby processing and presentation of dead cell-associated Ag. We have further revealed novel Clec9A interactions with ER-associated membrane proteins and chaperones that are differentially modulated in the presence of dead cells, shedding light on cDC1 handling of dead cell antigenic cargo.

## Discussion

Our research has identified a novel Clec9A interacting protein, the E3 ubiquitin ligase RNF41. We revealed that RNF41 regulates Clec9A levels, and thereby DAMP recognition and Ag-processing for presentation. We demonstrated that RNF41 directly binds to extracellular domains of Clec9A, and mediates Clec9A ubiquitination at membrane-associated cellular compartments, such as the ER and endosomal vesicles. This process is dependent on the RING domain function, and results in ubiquitination of several lysine residues in the stalk and CTLD domains of Clec9A; notably, the cytoplasmic regions of Clec9A are not ubiquitinated. Ubiquitination can then target Clec9A for degradation. This is consistent with the role of RNF41 in regulating the steady-state levels of other receptors, including ErbB3, IL3R and EPOR, in a ligand-independent manner, by direct binding, ubiquitination and targeting of these receptors for degradation (*Fry et al., 2011*; *Jing et al., 2008*; *Qiu and Goldberg, 2002*). However, this is the first description, to our knowledge, that the extracellular domain of a membrane receptor can interact with an E3 ubiquitin ligase within the cell, and be modified by ubiquitination for the regulation of receptor levels. This likely reflects the unique properties of Clec9A as a receptor specialized for the delivery of dead cell Ag to the endosomal compartments for cross-presentation.

We propose that at steady state, RNF41 acts to regulate Clec9A levels thereby limiting the Clec9A receptors available for dead cell Ag processing and presentation. We and others have found RNF41 to be associated with the ER and endosomes, as well as the cytosol (*Diamonti et al., 2002*; *Fry et al., 2011*; *Masschaele et al., 2017*). We demonstrated that Clec9A and RNF41 interact within steady- state DC, and that ubiquitinated Clec9A is present in membrane-associated cellular fractions that include the ER and recycling endosomes.

We propose that RNF41 can regulate Clec9A at two intracellular locations, at the ER and at endosomes. Our data indicates that RNF41 can interact with, and degrade newly synthesized Clec9A molecules at the ER as a means of controlling the abundance of Clec9A receptors. The increased association of Clec9A with ER molecules Erlin1/2 (SPFH1/2) in the presence of RNF41 supports this

model. The ER-Associated Degradation (ERAD) pathway plays an important role in maintaining abundance of membrane receptors (*Printsev et al., 2017*), and both RNF41 and Erlin1/2 have been implicated in this process: RNF41 can interact with newly synthesized ErbB3 and target it for degradation via ERAD as a means of controlling ErbB3 receptor levels (*Fry et al., 2011*); Erlin1/2 can recruit RNF170 to activated IP$_3$ receptors facilitating receptor ubiquitination and degradation via ERAD (*Lu et al., 2011*). Erlin1/2 have also been implicated in viral egress from the ER to cytosol (*Inoue and Tsai, 2017*). However, the topology of this interaction at the ER remains to be determined. While Clec9A and the Erlin1/2 extracellular domains are localized within the lumen of the ER, and RNF41 is associated with the ER membrane, further studies are required to determine whether RNF41 localises and interacts with Clec9A at the luminal or the cytoplasmic side of the ER membrane. We propose that the Clec9A-RNF41-Erlin interaction may facilitate translocation of Clec9A ectodomains or RNF41 across the ER membranes, thereby regulating access to ubiquitination and ERAD machinery. Further studies will be required to determine the molecular mechanisms that underpin these processes and the contribution of this pathway to the regulation of Clec9A fate.

Our data indicates that RNF41 also interacts with Clec9A at endosomes, and that this interaction is potentiated at lower pH as typically observed in late endosomes and lysosomes, rather than early endosomes. However, further experimental studies are required to identify the specific endosomal compartments at which the Clec9A-RNF41 interaction occurs. We demonstrated that Clec9A interactions with RNF41 and Erlin2, and Clec9A ubiquitination, were decreased at early stages following DC uptake of dead cells. This suggests that Clec9A interaction with dead cells may transiently divert Clec9A and its cargo away from RNF41-mediated degradation, to prolong Clec9A and cargo survival and thereby potentiate Ag presentation. How Clec9A on cDC1 delivers dead cell Ag for cross-presentation to CD8$^+$ T cells remains topical. Two major pathways have been proposed for cross-presentation: (i) a vacuolar pathway where Ag are processed by lysosomal proteases in the endosomes and loaded onto MHCI, and (ii) a cytosolic pathway, in which Ag are translocated to the cytosol for processing then back to the ER/endosomes for MHCI loading (*Gros and Amigorena, 2019*; *Grotzke and Cresswell, 2015*; *Grotzke et al., 2017b*). cDC1 cross-presentation of cell-associated Ag has been shown to occur through a Transporter-associated with Ag Processing (TAP)−1 dependent pathway for peptide translocation, implicating the cytosolic pathway (*Theisen et al., 2018*), although the relative contributions of the different proposed pathways for cross-presentation of dead cell Ag remains topical. Some ER proteins, and particular components of ERAD machinery, have also been implicated in cross-presentation (reviewed in *Gros and Amigorena, 2019*; *Grotzke and Cresswell, 2015*; *Grotzke et al., 2017b*). ER chaperones Calreticulin, Calnexin and translocon component Sec61 have been shown to be recruited to Ag-containing endosomes in bone marrow-derived DC for cross-presentation, although the direct contribution of Sec61 to cross-presentation has been challenging to assess (*Croce et al., 2017*; *Grotzke et al., 2017a*; *Guermonprez et al., 2003*; *Zehner et al., 2015*). The increased Clec9A-Calreticulin interaction following dead cell uptake, and the Clec9A-Calreticulin interaction with internalized dead cells supports a role for Calreticulin in chaperoning Clec9A or its cargo to facilitate cross-presentation; whether this requires translocation of Clec9A-Ag complexes across endosomal membranes for cytosolic processing or whether it occurs within the endosomes remains to be determined. However, the recent identification of active ubiquitination and proteasomes in endocytic compartments of DC and their contribution to Ag cross-presentation (*Sengupta et al., 2019*) provides an attractive model by which Clec9A and its dead cell cargo may access RNF41, ubiquitination and Ag-processing pathways within endosomes, potentially bypassing the need for translocation mediators. Determining the mechanism that enables RNF41 interaction with Clec9A at the endosomes, and the role of translocation mediators, remains an important issue for future studies.

The nature of the ubiquitin linkages on Clec9A could provide some clues to the control of Clec9A fate. RNF41-mediated K48-linked ubiquitination has been associated with proteasomal degradation (*Fry et al., 2011*), and K63-linked ubiquitination with non-proteasomal functions such as trafficking and activation (*Wang et al., 2009*). Our proteomic analysis of RNF41-mediated Clec9A ubiquitination identified usage of K11-, K48- and K63- linked ubiquitin. The identification of all three linkages may point to several processes occurring to Clec9A receptors at different subcellular locations such as ER versus endosomes, or may reflect heterotypic or branched ubiquitination. The role of such chains is still poorly understood, but it is clear that they act as a unique coding signal affecting recognition by reader proteins and altering the fate of the ubiquitinated target; heterotypic K11/K48-

linked ubiquitination can promote rapid proteasomal degradation (*Yau et al., 2017*) whereas K48/K63 branched ubiquitin chains can amplify NF-kB signaling (*Ohtake et al., 2016*).

The anti-Clec9A mAb that we previously generated (24/04-10B4) is particularly good at delivering Ag to cDC1 for cross-presentation. Given that it can limit access of RNF41 to Clec9A (*Figure 1—figure supplement 1H*), our data offer an explanation for its superior effects for vaccine delivery. We postulate that 24/04-10B4 retards the ubiquitination and degradation of Clec9A, thus enhancing cross-presentation of its cargo, similar to dead cells. With this in mind, an even more effective cross-priming Ag delivery vehicle may be designed, one that potentially completely prevents RNF41 binding to Clec9A.

While the major known pathways for ubiquitination-mediated regulation of cell surface receptor fate occur by ubiquitination of cytoplasmic receptor domains, there are precedents that ubiquitination can also occur in non-cytosolic compartments (*Sengupta et al., 2019*; *Siegelman et al., 1986*). However, the contribution and underlying mechanisms of such pathways to receptor regulation remain unknown. Our results demonstrate that RNF41 regulates the DAMP receptor Clec9A by ubiquitination of its extracellular domains, to regulate receptor fate and dead cell handling for cross-presentation. We further reveal novel Clec9A associations with ER proteins previously identified to play a role in cross-presentation, revealing insights into the differential handling of Clec9A at steady state and following uptake of dead cell cargo. Further studies on Clec9A interactions, their specific intracellular localization, and the precise form of the Clec9A ubiquitination under different conditions of DC activation and Ag handling are required to obtain a complete picture of the role of RNF41 in the regulation of Clec9A fate and function.

# Materials and methods

### Key resources table

| Reagent type (species) or resource | Designation | Source or reference | Identifiers | Additional information |
|---|---|---|---|---|
| Strain, strain background (*Escherichia coli*) | JM109 | Promega | Cat# L2001 | Competent cells |
| Strain, strain background (*Escherichia coli*) | BL21(DE3)pLysS | Promega | Cat# L1195 | Competent cells for protein expression |
| Strain, strain background (*Mus musculus*) | OT-I/Ly5.1 transgenic mice | Own colony (*Lahoud et al., 2011* doi: 10.4049/jimmunol.1101176) | | |
| Strain, strain background (*Mus musculus*) | OT-II/Ly5.1 transgenic mice | Own colony (*Lahoud et al., 2011* doi: 10.4049/jimmunol.1101176) | | |
| Strain, strain background (*Mus musculus*) | Clec9A$^{-/-}$ mice | Own colony (*Caminschi et al., 2012* doi: 10.1016/j.molimm.2011.11.008) | | |
| Cell line (*Homo sapiens*) | Freestyle 293F cells | Thermo Fisher | Cat# R79007 | |
| Cell line (*Mus musculus*) | MutuDC1940 | *Fuertes Marraco et al., 2012* doi: 10.3389/fimmu.2012.00331 | Gift from Prof. Acha-Orbea | |
| Cell line (*Cricetulus griseus*) | CHO-K1/ CHO-K1 cells expressing OVA-FLAG | *Caminschi et al., 2007* doi: 10.1084/jem.20071351 | | |

*Continued on next page*

*Continued*

| Reagent type (species) or resource | Designation | Source or reference | Identifiers | Additional information |
|---|---|---|---|---|
| Transfected construct (*Homo sapiens*) | FLAG-tagged full length human CLEC9A (hCLEC9A-FLAG) | *Caminschi et al., 2008* doi: 10.1182/blood-2008-05-155176 | | Construct to transiently transfect Freestyle 293F |
| Transfected construct (*Mus musculus*) | FLAG-tagged full length mouse Clec9A (mClec9A-FLAG) | *Caminschi et al., 2008* doi: 10.1182/blood-2008-05-155176 | | Construct to transiently transfect Freestyle 293F |
| Transfected construct (*Homo sapiens*) | FLAG-tagged recombinant human CLEC9A ecto-domains (hCLEC9A-ecto) | *Zhang et al., 2012* doi: 10.1016/j.immuni.2012.03.009 | | Construct to transiently transfect Freestyle 293F |
| Ttransfected construct (*Mus musculus*) | FLAG-tagged recombinant mouse Clec9A ecto-domains (mClec9A-ecto) | *Zhang et al., 2012* doi: 10.1016/j.immuni.2012.03.009 | | Construct to transiently transfect Freestyle 293F |
| Ttransfected construct (*Homo sapiens*) | RNF41 full-length mammalian expression construct in pcDNA3.1+ RNF41 region: 1-MGYD…VEEI-317 | This paper, synthesized by GeneArt, Life Technologies | | Construct to transiently transfect Freestyle 293F. Materials & Methods: Generation of Recombinant Proteins |
| Transfected construct (*Homo sapiens*) | RNF41- ΔRING mammalian expression construct in pcDNA3.1+ RNF41 region: 72-MRNM…VEEI-317 | This paper, synthesized by GeneArt, Life Technologies | | Construct to transiently transfect Freestyle 293F. Materials & methods: Generation of Recombinant Proteins |
| Antibody | anti-FLAG (clone M2, Mouse monoclonal, HRP-conjugate) | Sigma | Cat# A8592 | WB (1:6000) |
| Antibody | anti-FLAG, FITC (rat monoclonal) | WEHI Antibody Facility | clone 9H1 | FACS (1:200) |
| Antibody | anti-Myc (mouse monoclonal) | WEHI Antibody Facility | clone 9E10 | WB (1:3000) |
| Antibody | anti-RNF41 (rabbit polyclonal) | Sigma | Cat# HPA016812 | WB (1:3000) |
| Antibody | anti-RNF41 (rabbit polyclonal) | Bethyl Laboratories | Cat# A300-048A | WB (1:3000) |
| Antibody | anti-RNF41 (rabbit polyclonal) | Bethyl Laboratories | Cat# A300-049A | IF (1:50) PLA (1:100) |
| Antibody | anti-mouse Clec9A (rat monoclonal) | *Caminschi et al., 2008* doi: 10.1182/blood-2008-05-155176 | clone 24/04-10B4 | WB (2μg/ml) |
| Antibody | anti-mouse Clec9A (mouse polyclonal) | This paper | Used for confocal microscopy and PLA | IF (1:50) PLA (1:100) |
| Antibody | anti-Actin, HRP (goat polyclonal) | Santa Cruz | Cat# sc-1616HRP clone I-19 | WB (1:5000) |

*Continued on next page*

*Continued*

| Reagent type (species) or resource | Designation | Source or reference | Identifiers | Additional information |
|---|---|---|---|---|
| Antibody | anti-GAPDH, HRP (mouse monoclonal) | Thermo Fisher | Cat# MA5-15738-HRP Clone GA1R | WB (1:15000) |
| Antibody | anti-Ubiquitin, HRP (mouse monoclonal) | Santa Cruz | Cat# sc-8017 Clone P4D1 | WB (1:3000) |
| Antibody | anti-Ubiquitin (rabbit polyclonal) | Abcam | Cat#ab7780 | PLA (1:250) |
| Antibody | anti-K48 specific Ubiquitin (rabbit monoclonal) | Merck Millipore | Cat# 05-1307 Clone Apu2 | WB (1:3000) |
| Antibody | anti-Erlin1 (rabbit polyclonal) | Proteintech | Cat# 17311-1-AP | WB (1:3000) |
| Antibody | anti-Erlin2 (rabbit polyclonal) | Sigma | Cat # HPA002025 | WB (1:3000) IF (1:100) PLA (1:100) |
| Antibody | anti-Calreticulin (rabbit polyclonal) | Abcam | Cat#Ab2907 | WB (1:4000) IF (1:100) PLA (1:100) |
| Antibody | Anti-calnexin (rabbit polyclonal) | Abcam | Cat#ab22595 | PLA (1:250) |
| Antibody | anti-EEA-1 (mouse monoclonal) | Santa Cruz | Cat#sc-365652 Clone E-8 | WB (1:1500) IF (1:100) |
| Antibody | anti-Rab11 (mouse monoclonal) | BD Biosciences | Cat#610656 Clone 47 | WB (1:1000) IF (1:100) |
| Antibody | anti-mouse CD45.1, APC (mouse monoclonal) | BD Pharmingen | Cat# 558701 Clone A20 | FACS (1:200) |
| Antibody | anti-mouse CD45.2, Pacific Blue (mouse monoclonal) | BioLegend | Cat# 109820 Clone 104 | FACS (1:200) |
| Antibody | anti-mouse MHC II (hamster monoclonal) | WEHI | Clone N22 | IF (10µg/ml) |
| Antibody | anti-mouse CD8, PE (rat monoclonal) | WEHI | Clone YTS169.4 | FACS (1:200) |
| Antibody | Anti-mouse CD4 (rat monoclonal) | WEHI | Clone GK1.5 | FACS (1:400) |
| Antibody | anti-Rat Ig, HRP (goat polyclonal) | Southern Biotech | Cat# 3010-05 | WB (1:5000) |
| Antibody | anti-Rabbit IgG (H+L) Highly Cross-Adsorbed Secondary Antibody, Alexa Fluor 488 (donkey polyclonal) | Life Technologies | Cat#A-21206 | IF (1:200) |
| Antibody | anti-Rabbit IgG, HRP (goat polyclonal) | Bio-rad | Cat# 1706515 | WB (1:5000 - 1:10000) |
| Antibody | anti-Mouse IgG (H+L) Highly Cross-Adsorbed Secondary Antibody, Alexa Fluor 546 (goat polyclonal) | Life Technologies | Cat#A-11030 | IF (1:200) |

*Continued*

| Reagent type (species) or resource | Designation | Source or reference | Identifiers | Additional information |
|---|---|---|---|---|
| Antibody | anti-Mouse IgG (H+L) Cross-Adsorbed Secondary Antibody, Alexa Fluor 647 (goat polyclonal) | Life Technologies | Cat#A-21235 | IF (1:200) |
| Antibody | anti-Mouse IgG (H+L) Cross-Adsorbed Secondary Antibody, (Goat polyclonal, Alexa Fluor 488-conjugate) | Life Technologies | Cat#A-11029 | IF (1:200) |
| Recombinant DNA reagent | pLMP-Cherry (plasmid) | *Majewski et al., 2008* doi: 10.1371/journal.pbio.0060093 | | Knockdown plasmid for shRNA |
| Recombinant DNA reagent | Vsv-g (plasmid) | *Majewski et al., 2008* doi: 10.1371/journal.pbio.0060093 | | Envelope plasmid for shRNA knockdown |
| Recombinant DNA reagent | pMD1-gag-pol (plasmid) | *Majewski et al., 2008* doi: 10.1371/journal.pbio.0060093 | | Packaging plasmid for shRNA knockdown |
| Recombinant DNA reagent | RNF41-RBCC bacterial expression construct in pGex2T RNF41 region: 1-MGYD...VEEI-317 | This paper | | For recombinant protein production in bacteria. Materials & methods: Generation of Recombinant Proteins |
| Recombinant DNA reagent | RNF41-BCC bacterial expression construct in pGex2T RNF41 region: 72-MRNM...VEEI-317 | This paper | | For recombinant protein production in bacteria. Materials & methods: Generation of Recombinant Proteins |
| Recombinant DNA reagent | RNF41-CC bacterial expression construct in pGex2T RNF41 region: 135-IKHL...VEEI-317 | This paper | | For recombinant protein production in bacteria. Materials & methods: Generation of Recombinant Proteins |
| Recombinant DNA reagent | RNF41-C bacterial expression construct in pGex2T RNF41 region: 169-DIQL...VEEI-317 | This paper | | For recombinant protein production in bacteria. Materials & methods: Generation of Recombinant Proteins |
| Recombinant DNA reagent | RNF41-RBC bacterial expression construct in pGex2T RNF41 region: 1-MGYD...RAIR-181 | This paper | | For recombinant protein production in bacteria. Materials & methods: Generation of Recombinant Proteins |
| Sequence-based reagent | RNF41 shRNA#5_F | Integrated DNA Technologies | Oligonucleotides for generation for shRNA | TCGAGAAGGTATATTGCTGTTGACAGTGAGCGCTCCTTTGGTGTTGTTTGTTTATAGTGAAGCCACAGATGTATAAACAAACAACACCAAAGGAATGCCTACTGCCTCGG |

*Continued on next page*

*Continued*

| Reagent type (species) or resource | Designation | Source or reference | Identifiers | Additional information |
|---|---|---|---|---|
| Sequence-based reagent | RNF41 shRNA#5_R | Integrated DNA Technologies | Oligonucleotides for generation for shRNA | AATTCCGAGGCAGT AGGCATTCCTTTGGT GTTGTTTGTTTATAC ATCTGTGGCTTCACTAT AAACAAACAACACCAA AGGAGCGCTCACTGT CAACAGCAATATACCTTC |
| Sequence-based reagent | RNF41 shRNA#10_F | Integrated DNA Technologies | Oligonucleotides for generation for shRNA | TCGAGAAGGTATATTG CTGTTGACAGTGAGCG ACTGGAGATGCCCAAA GATGAATAGTGAAGCC ACAGATGTATTCATCTT TGGGCATCTCCAGG TGCCTACTGCCTCGG |
| Sequence-based reagent | RNF41 shRNA#10_R | Integrated DNA Technologies | Oligonucleotides for generation for shRNA | AATTCCGAGGCAG TAGGCACCTGGAG ATGCCCAAAGATGA ATACATCTGTGGCT TCACTATTCATCTTTG GGCATCTCCAGTCGCT CACTGTCAACAGCAA TATACCTTC |
| Sequence-based reagent | RNF41-q14_F | Geneworks | qPCR primers | GTGAGCACAA CCCGAAGC |
| Sequence-based reagent | RNF41-q15_R | Geneworks | qPCR primers | GTTCATCTTTG GGCATCTCC |
| Peptide, recombinant protein | MHC class I (257-264) restricted OVA peptide | Mimotopes | | Positive control for Ag presentation to CD8+ T cells |
| Peptide, recombinant protein | MHC class II (323-339) restricted OVA peptide | Mimotopes | | Positive control for Ag presentation to CD4+ T cells |
| Peptide, recombinant protein | Mouse IgGκ binding protein, HRP | Santa Cruz | Cat#sc-516102 | WB (1:5000) |
| Peptide, recombinant protein | Streptavidin-HRP | GE Healthcare | Cat# RPN1231 | ELISA (1:15 000) |
| Peptide, recombinant protein | Streptavidin-PE | BD Biosciences | Cat# 554061 | FACS (1:200) |
| Commercial assay or kit | West Pico Super Signal ECL substrate | Thermo Fisher | Cat# 34080 | |
| Commercial assay or kit | Luminata Forte Western HRP substrate | Merck Millipore | Cat# WBLUF0500 | |
| Commercial assay or kit | Pierce BCA Protein Assay kit | Thermo Fisher | Cat# 23225 | |
| Commercial assay or kit | Ubiquitinylation kit | Enzo Life | Cat# BML-UW9920 | |
| Commercial assay or kit | BirA | Avidity | Cat# bulk BirA | |

*Continued on next page*

*Continued*

| Reagent type (species) or resource | Designation | Source or reference | Identifiers | Additional information |
|---|---|---|---|---|
| Commercial assay or kit | ProtoArray Human Protein MicroArray Version 4.1 PPI Kit for biotinylated proteins | Thermo Fisher | Cat# PAH05241011 | |
| Commercial assay or kit | PKH26 labeling | Sigma-Aldrich | Cat#MINI26-1KT | |
| Commercial assay or kit | Duolink In Situ PLA Probe Anti-Rabbit PLUS | Sigma-Aldrich | Cat#DUO92002 | |
| Commercial assay or kit | Duolink In Situ PLA Probe Anti-Mouse MINUS | Sigma-Aldrich | Cat#DUO92004 | |
| Commercial assay or kit | Duolink In Situ Detection Reagents Orange | Sigma-Aldrich | Cat#DUO92007 | |
| Commercial assay or kit | Duolink In Situ Detection Reagents FarRed | Sigma-Aldrich | Cat#DUO92013 | |
| Commercial assay or kit | Duolink In Situ Detection Reagents Green | Sigma-Aldrich | Cat#DUO92014 | |
| Chemical compound, drug | MG132 in Solution | Calbiochem | Cat# 474791 | |
| Chemical compound, drug | PR619 | Life Sensors | Cat# SI9619 | |
| Chemical compound, drug | Brefeldin A | Sigma-Aldrich | Cat#B5936-200UL | |
| Software, algorithm | Prospector v5 (Thermo Fisher) | https://www.thermofisher.com/au/en/home/life-science/protein-biology/protein-assays-analysis/protein-microarrays/technical-resources/data-analysis.html | | |
| Software, algorithm | Image Lab v5.2.1 (Bio-rad) | http://www.bio-rad.com/en-au/product/image-lab-software?ID=KRE6P5E8Z#fragment-6 | Image Lab Software, RRID:SCR_014210 | |
| Software, algorithm | FIJI/ImageJ version 2.0.0 | https://fiji.sc | ImageJ, RRID:SCR_003070 | |
| Software, algorithm | FlowJo 10.5.0 (Treestar Inc) | https://www.flowjo.com/solutions/flowjo/downloads | FlowJo, RRID:SCR_008520 | |
| Software, algorithm | Rotor-gene Q series software (Qiagen) | https://www.qiagen.com/au/resources/resourcedetail?id=9d8bda8e-1fd7-4519-a1ff-b60bba526b57&lang=en | Rotor-Gene Q Series Software, RRID:SCR_015740 | |

*Continued*

| Reagent type (species) or resource | Designation | Source or reference | Identifiers | Additional information |
|---|---|---|---|---|
| Software, algorithm | Weasel v3.0.2 | http://www.frankbattye.com.au/Weasel/ | | |
| Software, algorithm | GraphPad Prism 7.0b | https://www.graphpad.com/scientific-software/prism/ | GraphPad Prism, RRID:SCR_002798 | |
| Software, algorithm | Byonic (Protein Metrics) | https://www.proteinmetrics.com/products/byonic/ | PMI-Byonic, RRID:SCR_016735 | |
| Software, algorithm | MaxQuant v1.6.0.16 | https://www.biochem.mpg.de/5111795/maxquant | MaxQuant, RRID:SCR_014485 | |
| Software, algorithm | Perseus v1.6.1.3 | https://www.biochem.mpg.de/5111810/perseus | Perseus, RRID:SCR_015753 | |
| Other | Bond-Elut Omix-mini Bed 96 C18 | Agilent Technologies | Cat# A57003MBK | |
| Other | Anti-FLAG M2 affinity gel | Sigma-Aldrich | Cat# A2220 | |
| Other | TUBE-agarose | Life Sensors | Cat# UM401 | |
| Other | Glutathione Sepharose 4B | GE Healthcare | Cat# 17-0756-05 | |
| Other | Protein G Sepharose | GE Healthcare | Cat# 17061801 | |
| Other | PNGase F | New England Biolabs | Cat# P0704S | |
| Other | Endo H | New England Biolabs | Cat# P0702S | |
| Other | 293fectin transfection reagent | Thermo Fisher | Cat# 12347500 | |
| Other | Freestyle MAX transfection reagent | Thermo Fisher | Cat# 16447100 | |

## Mice

C57BL/6J, *Clec9a*$^{-/-}$ (*Caminschi et al., 2012*) and OT-I and OT-II (*Caminschi et al., 2008*; *Lahoud et al., 2011*) mice were bred and maintained under specific pathogen-free (SPF) conditions at Alfred Medical Research and Education Precinct or Monash Animal Research Platform. C57BL/6J and *Clec9a*$^{-/-}$ mice were used at 6–12 weeks of age, were age-matched and gender-matched, and used for generation of WT and *Clec9a*$^{-/-}$ FLt3L-DC. OT-I and OT-II mice were used at 9–15 weeks of age, for the isolation of transgenic T cells. All experiments were conducted under the approval of the Alfred Medical Research and Education Precinct Animal Ethics Committees (Application #2888) or the Monash Animal Research Platform Animal Ethics Committee (MARP/2015/096; 21639), Monash University, Clayton, Australia and in accordance with National Health and Medical Research Council (NHMRC) Australia guidelines.

## Cell lines

Freestyle 293 F cells were purchased from Thermo Fisher, and were maintained in Freestyle 293 Expression media (Thermo Fisher) in a humidified $CO_2$ incubator at 37°C, 8% $CO_2$ and on an orbital shaking platform at 150 rpm. MutuDC 1940 were kindly provided by Prof. Acha-Orbea who generated the cell line. MutuDC 1940 were cultured at 37°C, 10% $CO_2$ in IMDM (Gibco), 10% FCS (In Vitro

Technologies), 100 U/ml penicillin, 100 μg/ml streptomycin (Sigma-Aldrich), 27.5 μM 2-Mercaptoethanol (Gibco) and osmolarity adjusted to 308 mOsm. CHO-K1 and CHO-OVA cells were maintained in RPMI-1640 (Gibco), HEPES (Gibco), 5% FCS, 100 U/ml penicillin, 100 μg/ml streptomycin at 10% $CO_2$, 37°C. CHO-OVA were cultured in the presence of 0.5 mg/ml G418 (Astral Scientific). All cell lines were routinely tested for mycoplasma contamination and found to be negative.

## Isolation of spleen conventional DC

Spleen conventional DC (cDC) were isolated from C57BL/6 wild-type (WT) or *Clec9a*[-/-] mice, as previously described (*Caminschi et al., 2008*) and cDC1 further enriched by depletion using mAb (M1/70 anti-CD11b, p84 anti-Sirpα, 120G8 anti-BST-2 and F4/80) and anti-rat Ig magnetic beads (Dynabeads, Invitrogen).

## Differentiation of bone-marrow Flt3L-derived dendritic cells

Primary bone marrow Flt3L-derived DC (Flt3L-DC) were generated by culturing bone marrow from wild type (C57BL/6J) or *Clec9a*[-/-] mice in complete media: RPMI-1640 supplemented with 1 mM sodium pyruvate (Sigma-Aldrich), 10 mM HEPES, 25 mM sodium carbonate (Merck Millipore), 100 U/ml penicillin, 100 μg/ml streptomycin, 2 nM L-glutamine, 50 μM 2-mercaptoethanol and 10% FCS. Cells were cultured with 200–400 ng/ml Flt3L (In house; Bio X Cell) and 300 pg/ml GM-CSF (Peprotech). On day six the culture was supplemented with 1 ng/ml GM-CSF and DC were harvested on day 8.

## Generation of recombinant proteins

Clec9A ecto-domains and control proteins fused to a FLAG-tag and a biotinylation consensus sequence were expressed in mammalian Freestyle 293 F cells (Thermo Fisher) using Freestyle Max (Thermo Fisher) and purified using anti-FLAG M2 affinity gel (Sigma-Aldrich) and size exclusion chromatography (SEC) as previously described (*Zhang et al., 2012*). cDNA constructs encoding full-length FLAG-tagged Clec9A were subcloned into either pEF-Bos or pcDNA3.1+ (Invitrogen) for mammalian expression as previously described (*Zhang et al., 2012*). cDNA constructs encoding full-length and truncated versions of mouse and human RNF41 for mammalian expression (RNF41 full-length: 1-MGYD...VEEI-317; RNF41- ΔRING: 72-MRNM...VEEI-317) were subcloned into pcDNA3.1 + (Invitrogen). cDNA constructs encoding RNF41 for bacterial expression (RNF41-RBCC: 1-MGYD...VEEI-317; RNF41-BCC: 72-MRNM...VEEI-317; RNF41-CC: 135-IKHL...VEEI-317; RNF41-C: 169-DIQL...VEEI-317; RNF41-RBC: 1-MGYD...RAIR-181) were cloned into a modified pGEX-2T vector (GE Healthcare). His-tagged and untagged RNF41 proteins were expressed in mammalian Freestyle 293 F cells using 293Fectin (Thermo Fisher). GST-fusion RNF41 recombinant proteins were expressed in bacterial BL21 DE3 *E. coli* (Promega) and purified using Glutathione-Sepharose resin (GE Healthcare) and SEC.

## Identification of Clec9A interacting proteins

Mouse Clec9A-ecto and Cire-ecto were enzymatically biotinylated using BirA (Avidity) as previously described (*Zhang et al., 2012*). Biotinylated proteins were hybridized to protein arrays (ProtoArray Human Protein MicroArray Version 4.1 PPI Kit for biotinylated proteins, Catalog no. PAH05241011, Lot number HA20139, Thermo Fisher) and bound proteins detected using Streptavidin-Alexa Fluor 647 as per manufacturer's instructions. Data was collected using a GenePix 4000B (Molecular Devices) using an excitation of 635 nm and analyzed using Prospector v5 (Thermo Fisher) for specific mClec9A binding hits (Z-score >3).

## Protein-protein interaction by the pull-down assay

Glutathione-sepharose resin coupled with either GST or GST-RNF41 proteins was resuspended in 0.2% NP-40 and 2% glycerol in 20 mM Tris-buffered saline, pH 7.5 containing Complete protease inhibitor cocktail (Roche) and 1 mM PMSF, incubated with 1 μg/ml purified FLAG-tagged soluble Clec9A and incubated at 4°C for 2 hr. Unbound proteins were washed away. Bound proteins were eluted from the beads using 2 x SDS reducing sample at 92°C and analyzed by western blot with anti-FLAG antibody (Sigma-Aldrich).

## Western blot analysis

Protein concentrations from cell lysates were normalized using Pierce BCA Protein Assay kit (Thermo Fisher) and electrophoresis performed using NuPAGE 4–12% Bis-Tris gradient gels (Invitrogen). Resolved proteins were transferred to Immobilon PVDF membrane (Merck Millipore) using Xcell II blot module. Membranes were blocked with 5% skim milk before the addition of immunoblotting antibodies. Immunoblots were developed using West Pico Super Signal Plus ECL substrate (Thermo Fisher) or Luminata Forte Western HRP substrate (Merck Millipore). Images were captured using a ChemiDoc Touch Imaging System (Bio-rad), and analyzed using Image Lab Software (Bio-rad). Clec9A, Ubiquitin, K-48 Ubiquitin, Rab11, Actin and Gapdh were analyzed under non-reducing conditions unless otherwise indicated in figure legends. RNF41, Calreticulin, Erlin1, Erlin2 and EEA1 were analyzed under reducing condition.

## ELISA

ELISA plates (Costar) were coated with recombinant proteins (10 µg/ml), as indicated in the figure legends, blocked with 1% BSA then incubated with test proteins including FLAG-tagged Clec ecto-domain fragments, GST-RNF41 proteins or controls. Binding was detected with anti-FLAG HRP or rabbit anti-RNF41 Ab (Bethyl Laboratories, A300-049A) and anti-rabbit Ig HRP (Bio-Rad). Statistical analysis of ELISA data was performed by an unpaired two-tailed t-test on log-transformed data.

To measure the effect of pH on Clec9A binding to RNF41, ELISA plates were coated with GST-tagged RNF41 proteins (10 µg/ml), control GST (10 µg/ml), or actin complexes (platelet actin pre-incubated with GST-tagged βII-spectrin actin-binding domain (ABD): 10 µg/ml actin; 2.5 µg/ml ABD), blocked then incubated with biotinylated FLAG-tagged recombinant Clec9A-ectodomain proteins and controls in pH 4–8 adjusted blocking buffer. Binding was detected with streptavidin-HRP (GE Healthcare) in pH-adjusted blocking buffer.

## Dead cell uptake assays

MutuDC 1940 were cultured at 37°C for 2–18 hr with unlabeled or PKH26 (Sigma-Aldrich)-labeled CHO-K1 cells, that had undergone five freeze-thaw cycles as a source of dead cells. For the last hour of culture, DC were allowed to adhere to anti-MHCII (N22) pre-coated 6 mm wells on Teflon printed slides (ProSciTech) before fixation as previously described (*Chiang et al., 2016*). DC were subsequently stained with Ab for confocal analysis, or analyzed using a Proximity Ligation Assay (PLA). Each experiment included a dead cell uptake control; a sample of DC was cultured with PKH26-labeled CHO-K1 dead cells and analyzed for the percentage of DC that had taken up dead cells; over 96% of DC demonstrated dead cell uptake in all experiments.

## Confocal staining and analysis

Rabbit polyclonal anti-RNF41 antibody (Bethyl Laboratories, A300-049A) was validated for confocal microscopy by transfecting RNF41 into 293 F cells. Endogenous RNF41 was detected in untransfected 293 F cells, and there was an increase in RNF41 expression detected in 293 F cells transfected with full-length RNF41 (*Figure 2—figure supplement 1A*). No staining was detected using the secondary anti-rabbit Alexa Fluor 488 alone. Mouse polyclonal anti-mouse Clec9A antibodies (#312, #1203) for confocal microscopy were developed by immunizing *Clec9a⁻/⁻* mice (*Caminschi et al., 2012*) three times with 37–70 µg mouse Clec9A-ectodomain. Unimmunized mouse serum was used as a control. Anti-Clec9A polyclonal antibodies were validated by confocal microscopy. Clec9A expression was detected in 293 F cells transfected with Clec9A, but not in untransfected 293 F cells (*Figure 2—figure supplement 1B*) and in Flt3L-derived DC isolated from wild-type C57BL/6 mice but not from *Clec9a⁻/⁻* mice (*Figure 2—figure supplement 1C*).

For confocal analysis of DC, DC were allowed to adhere to anti-MHCII (N22) pre-coated 6 mm wells on Teflon printed slides (ProSciTech) for 1 hr, washed with PBS and fixed with 4% paraformaldehyde (Sigma-Aldrich). After fixation, cells were permeabilized with 0.3% Triton X-100 (Sigma-Aldrich)/PBS for 5 min and blocked with 10% normal goat serum (Life technologies)/0.1% Triton X-100/PBS (blocking buffer) for 1 hr. Slides were incubated with anti-EEA1 (mouse clone E-8, Santa Cruz), anti-Rab11 (mouse polyclonal, BD Biosciences), anti-RNF41 (049A, Bethyl Laboratories) or anti-Clec9A polyclonal mouse serum in blocking buffer, followed by washing and incubation with anti-rabbit H+L Alexa Fluor 488 and anti-mouse H+L Alexa Fluor 647 (both Life technologies). Cells

were stained with DAPI (Sigma-Aldrich) and mounted using DAKO mounting media (DAKO) with 100 mg/ml DABCO anti-fade reagent (Sigma-Aldrich).

The proximity ligation assays (PLA) were performed using Duolink technology (Sigma-Aldrich) as per manufacturers' instructions. Cells were fixed and permeabilized as above, and stained using the mouse anti-Clec9A serum and rabbit anti-RNF41 A300-049A, rabbit anti-Calreticulin, rabbit anti-Calnexin, rabbit anti-Erlin2, or rabbit anti-Ubiquitin as primary antibodies. These were detected using Duolink In Situ PLA Probe Anti-Mouse MINUS Affinity purified Donkey anti-Mouse IgG (H+L) and Duolink In Situ PLA Probe Anti-Rabbit PLUS Affinity purified Donkey anti-rabbit IgG (H+L) respectively. Proximity ligation was detected using the Green, Orange or FarRed Duolink In Situ Detection reagents and cells were mounted using Duolink In Situ Mounting Medium with DAPI (all Sigma-Aldrich).

All images were acquired on Nikon Ar1 and C1 confocal microscopes with a 1.4 NA 60x oil immersion objective and 405, 488, 561, 638 nm lasers, using NIS-Elements imaging software. Acquired images were analyzed with ImageJ2.0.0 version software (Wayne Rasband, National Institutes of Health, USA). For image analysis, colocalization images were generated using Image J software, which only displays pixels that have a positive signal in both channels of interest (eg. PLA: Calreticulin+Clec9A and PKH26-labeled dead cells in *Figure 5E*) with a fluorescence intensity above the threshold set using negative controls. Double positive pixels, determined by ImageJ, are displayed in white showing colocalization between the two proteins of interest, whereas single positive and negative pixels are shown in black. For quantification studies, colocalization was assessed by Manders' coefficients (*Chiang et al., 2016*; *Hutten et al., 2016*), which determines the fraction of pixels that have positive signals for both channels of interest above an arbitrary threshold set using negative controls, with the JACoP plugin toolbox (*Bolte and Cordelières, 2006*). The PLA fluorescence and expression of Clec9A or RNF41 (*Figure 5—figure supplement 1*) was quantitated by thresholding to remove background and using the particle analysis tool in Image J to calculate the area of PLA or expression per cell. Data was analyzed by students t-test, Mann– Whitney test, one-way ANOVA and Tukey's multiple comparisons test or Kruskal-Wallis with Dunn's multiple comparisons, as described in figure legends.

## Immuno-electron microscopy

Immuno-electron microscopy on Tokuyasu cryo-sections was performed as described in *Slot and Geuze, 2007*. DC were fixed in 4% paraformaldehyde (Electron Microscopy Sciences) in 0.1M phosphate buffer (pH7.4) overnight at 4°C and then stored in 1% paraformaldehyde. Following rinses in PBS and then 0.15% glycine/PBS, DC were scraped from petri dishes in a 1% porcine skin gelatine (Sigma)/PBS solution, centrifuged, and pellets resuspended in 12% porcine skin gelatine/PBS for 30 min at 37°C. DC were pelleted, cooled at 4°C for 30 min, cut into 1 mm$^3$ blocks on ice and infiltrated in 2.3M sucrose overnight at 4°C. Cell blocks were mounted on bullseye cryo-pins (Electron Microscopy Sciences) using 2.3M sucrose and frozen in liquid nitrogen before cryo-sectioning using a Leica FC7/UC7 cryo-ultramicrotome. Ultra-thin cryo-sections were retrieved using a 2% methylcellulose/2.3M sucrose solution and stored on formvar coated grids at 4°C prior to immunolabeling.

Sections on grids were rinsed in PBS at 37°C for 1 hr to remove gelatine and aldehydes inactivated using 0.15% glycine/PBS. After blocking in 1%BSA/PBS (blocking buffer), sections were incubated with anti-RNF41 rabbit polyclonal (Bethyl Laboratories, Cat# A300-049A) in blocking buffer, washed in 0.1%BSA/PBS (wash buffer) and incubated with 10 nm Protein-A-Gold (Utrecht University) in blocking buffer, washed again in wash buffer and PBS and fixed in 1% glutaraldehyde (PolySciences). Following washes in water, sections were contrasted in 2% uranyloxaloacetate (pH7) and then 2% methylcellulose/4% uranylacetate (pH4) on ice, before drying and imaging using a JEOL-1400Plus transmission electron microscope at 80keV.

## Retrovirus production and shRNA-mediated knockdown

Constructs encoding RNF41 were generated by annealing forward and reverse oligos (RNF41 shRNA#5 forward: 5'-TCGAGAAGGTATATTGCTGTTGACAGTGAGCGCTCCTTTGGTGTTGTTTG TTTATAGTGAAGCCACAGATGTATAAACAAACAACACCAAAGGAATGCCTACTGCCTCGG-3'; RNF41 shRNA#5 reverse: 5'-AATTCCGAGGCAGTAGGCATTCCTTTGGTGTTGTTTGTTTATACATC TGTGGCTTCACTATAAACAAACAACACCAAAGGAGCGCTCACTGTCAACAGCAATATACCTTC-3';

RNF41 shRNA#10 forward: 5'-TCGAGAAGGTATATTGCTGTTGACAGTGAGCGACTGGAGA TGCCCAAAGATGAATAGTGAAGCCACAGATGTATTCATCTTTGGGCATCTCCAGGTGCCTAC TGCCTCGG-3'; RNF41 shRNA#10 reverse: 5'-AATTCCGAGGCAGTAGGCACCTGGAGA TGCCCAAAGATGAATACATCTGTGGCTTCACTATTCATCTTTGGGCATCTCCAGTCGCTCACTG TCAACAGCAATATACCTTC-3'), and cloning into an LMP-Cherry knockdown vector, to facilitate Cherry based selection of transduced cells. Retrovirus was generated by incubating 293 T cells with 25 μM chloroquine followed by transfection with the LMP-Cherry-RNF41 knockdown plasmid together with Envelope plasmid (VSV-g) and Packaging plasmid (pMD1-gag-pol) using the calcium phosphate precipitation method (*Majewski et al., 2008*) and virus containing supernatants harvested. MutuDC 1940 were virally transduced with retrovirus encoding RNF41 shRNA in the presence of 4 μg/ml polybrene and centrifugation at 1000 x *g*. The following day the DC were cultured in fresh IMDM with 10% FCS and 2-Mercaptoethanol. Four days post-spinfection, the DC were harvested and virus-transduced cells sorted based on high level of mCherry expression using a FACSA-RIA (BD Biosciences). RNF41 knockdown was confirmed by RNF41 qPCR for RNA expression (Forward Oligo: RNF41-q14: GTGAGCACAACCCGAAGC; Reverse Oligo RNF41-q15: GTTCATC TTTGGGCATCTCC) using SYBR Green detection (Quantinova SYBR Green PCR Kit Qiagen #208054; Rotor-gene Q RT-PCR machine; Rotor-gene Q series software) and by western blot for protein expression.

## Ag presentation to CD8$^+$ and CD4$^+$ T cells in vitro

MutuDC 1940 cells transduced with RNF41 knockdown (#5 and #10) or non-silencing constructs (5 × 10$^4$ cells per well) were cultured with freeze-thawed CHO-K1 cells expressing OVA (*Caminschi et al., 2007*) as a source of dead cell-associated Ag, and CFSE-labeled purified OVA specific transgenic OT-I$^{Ly5.1}$ or OT-II $^{Ly5.1}$ cells (*Caminschi et al., 2007*) at a ratio of 1 DC: 1 T cell: two dead cells. Three days later, the proliferated OT-I and OT-II cells were enumerated by flow cytometry as previously described (*Caminschi et al., 2007*) and the percentage (%) increase of T cell proliferation calculated relative to the DC transduced with non-silencing shRNA (non-silenced group = 100%). As a positive control, MutuDC 1940 cells and T cells were cultured in the presence of 0.5 μg/ml MHC class I (257-264) restricted OVA peptide or MHC class II (323-339) restricted OVA peptide.

## Analysis of Clec9A interactions in a transient transfection system

FLAG-tagged full-length mouse Clec9A (mClec9A-FLAG), human CLEC9A (hCLEC9A-FLAG) or mouse Clec12A (mClec12A), RNF41, RNF41-ΔRING or Myc-tagged ubiquitin (Ub-Myc) were expressed in Freestyle 293 F cells by co-transfection of 10 μg of each expression plasmid or vector alone (pcDNA3.1+), as indicated, to a total of 30 μg plasmid using 293Fectin transfection reagent. Transfected cells were harvested 1–2 days post-transfection for analysis by flow cytometric analysis using a FACS CantoII (BD Biosciences) and WEASEL flow cytometry software, or by western blot analysis or immunoprecipitation studies.

## Immunoprecipitation studies

For immunoprecipitation of Clec9A from transfected cells: Transfected cells were lysed using 1% Triton X-100, 150 mM NaCl, 1 mM EDTA, 50 mM Tris at pH 6.0–7.5, supplemented with 50 μM PR619, Benzonase nuclease (Novagen) and Complete protease inhibitor cocktail (Roche). Lysates were pre-cleared with protein G Sepharose (GE Healthcare), and FLAG-tagged Clec9A immunoprecipitated using anti-FLAG M2 affinity gel (Sigma-Aldrich). Immunoprecipitated Clec9A was further analyzed by western blot or LC-MS/MS.

For analysis of Clec9A ubiquitination from DC fractions: MutuDC1940 were lysed in 25 μg/ml digitonin, 50 mM HEPES, 150 mM NaCl, 50 μM PR619, supplemented with Complete protease inhibitor cocktail (Roche). Lysates were subjected to centrifugation (4000 x *g* for 10 min) for separation of cytosol and early endosome fractions (supernatant; labeled as Digitonin Soluble) and cellular organelle/membrane fractions (pellet). Cellular organelles were subsequently solubilised using 1% IGE-PAL-CA 630, and subjected to further centrifugation (7000 x *g* for 10 min) to collect organelle lysates (supernatant; labeled as IGEPAL Soluble) and detergent-resistant membrane complexes (pellet; labeled IGEPAL Resistant). As a control, Total lysates were generated as described above (1%

Triton X-100, 150 mM NaCl, 1 mM EDTA, 50 mM Tris at pH 6.0, supplemented with Benzonase nuclease (Novagen) and Complete protease inhibitor cocktail (Roche)). Ubiquitinated proteins were immunoprecipitated from lysates (Total, Digitonin Soluble, IGEPAL Soluble Fractions) using Tandem Ubiquitin Binding Entity (TUBE) agarose (Life Sensors) and further analyzed by western blot under non-reducing conditions for Clec9A (anti-Clec9A clone 10B4) and ubiquitin.

## LC-MS/MS analysis

Clec9A interacting proteins were immunoprecipitated from cells transfected with Clec9A +/− RNF41 under 1% Triton X-100 detergent lysis and wash conditions (1% Triton X-100, 150 mM NaCl, 1 mM EDTA, 50 mM) Tris at pH 6.0 supplemented with 50 µM PR619, Benzonase nuclease (Novagen) and Complete protease inhibitor cocktail (Roche) as described above. Immunoprecipitated proteins for LC-MS/MS analysis were eluted with 0.2 M glycine pH 2.5 and neutralized with 1 M Tris pH 8.0. Proteins were reduced with 5 mM TCEP (Thermo Fisher), alkylated with 50 mM iodoacetamide (Sigma-Aldrich), and digested with sequencing grade trypsin (Promega). Samples were acidified to 1% formic acid and purified with Bond-Elut OMIX C18 tips (Agilent) prior to mass spectrometry.

Using a Dionex UltiMate 3000 RSLCnano system equipped with a Dionex UltiMate 3000 RS autosampler, the samples were loaded via an Acclaim PepMap 100 trap column (100 µm x 2 cm, nanoViper, C18, 5 µm, 100 å; Thermo Scientific) onto an Acclaim PepMap RSLC analytical column (75 µm x 50 cm, nanoViper, C18, 2 µm, 100 å; Thermo Scientific). The peptides were separated by increasing concentrations of 80% ACN/0.1% FA at a flow of 250 nl/min for 128 min and analyzed with an Orbitrap Fusion Tribrid mass spectrometer (Thermo Scientific). Each cycle was set to a fixed cycle time of 2 s consisting of an Orbitrap full ms1 scan (resolution: 120.000; AGC target: 1e6; maximum IT: 54 ms; scan range: 375–1800 m/z) followed by several Orbitrap ms2 scans (resolution: 30.000; AGC target: 2e5; maximum IT: 54 ms; isolation window: 1.4 m/z; HCD Collision Energy: 32%). To minimize repeated sequencing of the same peptides, the dynamic exclusion was set to 15 s and the 'exclude isotopes' option was activated.

Acquired .raw files were searched against the human UniProtKB/SwissProt database (v2017_07) appended with the mouse Clec9A sequence using either MaxQuant (*Cox and Mann, 2008*) or Byonic (Protein Metrics) considering a false discovery rate (FDR) of 1% using the target-decoy approach. Carbamidomethylation was specified as a fixed modification. Oxidation of methionine, acetylation of protein N-termini and the Gly-Gly ubiquitination footprint motif on lysine residues were set as variable modifications. N-linked glycan modifications were identified in Byonic (Protein Metrics) using the implemented glycan databases. Trypsin was used as the enzymatic protease and up to two missed cleavages were allowed. Data visualization and mining was performed in Perseus (*Tyanova et al., 2016*) or Excel. Data was deposited to the ProteomeXchange consortium via the PRIDE (*Perez-Riverol et al., 2019*) partner repository.

## Statistical analysis

Data were first assessed for normal distribution, then analyzed for statistical significance by parametric and non-parametric tests as indicated in the figure legends. The number of biological repeats for each experiment is also indicated in the figure legends. $p < 0.05$ was considered significant and is represented as *$p < 0.05$; **$p < 0.01$; ***$p < 0.001$; ****$p < 0.0001$.

## Acknowledgements

The authors acknowledge the facilities, scientific and technical assistance of Monash Animal Research Platform, Monash Micro Imaging, Monash Flow Core, Monash Biomedical Proteomics Facility and the Monash Antibody Technology Facility, Monash University, Victoria, Australia. We thank Soo San Wan for technical assistance, Paul Masendycz, Kaye Wycherley and the WEHI Antibody Facility for assistance in developing Clec9A antibodies, the WEHI Flow Cytometry Laboratory, and the Australian Genome Research Facility. We thank Hans Acha Orbea, Jessica Dunleavy, Stephen Firth, Sandra Nicholson and Moira O'Bryan for their contribution of reagents and advice.

# Additional information

## Funding

| Funder | Author |
|---|---|
| National Health and Medical Research Council | Peter E Czabotar<br>Linda M Wakim<br>Justine D Mintern<br>Kristen Radford<br>Irina Caminschi<br>Meredith O'Keeffe<br>Jose Villadangos<br>Mark Wright<br>Marnie E Blewitt<br>William Heath<br>Ken Shortman<br>Anthony W Purcell<br>Nicos A Nicola<br>Jian-Guo Zhang<br>Mireille H Lahoud |
| Cancer Council Victoria | Kirsteen M Tullett |

The funders had no role in study design, data collection and interpretation, or the decision to submit the work for publication.

## Author contributions

Kirsteen M Tullett, Formal analysis, Validation, Investigation, Methodology, Writing - original draft, Writing - review and editing; Peck Szee Tan, Hae-Young Park, Ralf B Schittenhelm, Formal analysis, Investigation, Writing - original draft, Writing - review and editing; Nicole Michael, Antonia N Policheni, Emily Gruber, Cheng Huang, Alex J Fulcher, Formal analysis, Investigation; Rong Li, Formal analysis, Investigation, Writing - original draft; Jillian C Danne, Investigation; Peter E Czabotar, Linda M Wakim, Conceptualization, Supervision, Investigation; Justine D Mintern, Conceptualization, Supervision; Georg Ramm, Conceptualization, Formal analysis, Supervision, Investigation; Kristen J Radford, Meredith O'Keeffe, Jose A Villadangos, Mark D Wright, Anthony W Purcell, Conceptualization, Supervision, Writing - review and editing; Irina Caminschi, Conceptualization, Formal analysis, Supervision, Investigation, Writing - review and editing; Marnie E Blewitt, Conceptualization, Supervision, Investigation, Methodology, Writing - review and editing; William R Heath, Supervision, Writing - review and editing; Ken Shortman, Conceptualization, Funding acquisition, Investigation, Methodology, Writing - review and editing; Nicos A Nicola, Conceptualization, Supervision, Funding acquisition, Writing - review and editing; Jian-Guo Zhang, Conceptualization, Supervision, Investigation, Writing - review and editing; Mireille H Lahoud, Conceptualization, Resources, Data curation, Formal analysis, Supervision, Funding acquisition, Validation, Investigation, Methodology, Writing - original draft, Project administration, Writing - review and editing

## Author ORCIDs

Kirsteen M Tullett  http://orcid.org/0000-0002-2389-9819
Peck Szee Tan  http://orcid.org/0000-0002-8808-7125
Marnie E Blewitt  http://orcid.org/0000-0002-2984-1474
Mireille H Lahoud  https://orcid.org/0000-0001-8472-6201

## Ethics

Animal experimentation: All experiments were conducted under the approval of the Alfred Medical Research and Education Precinct Animal Ethics Committees (Application #2888) or the Monash Animal Research Platform Animal Ethics Committee (MARP/2015/096; 21639), Monash University, Clayton, Australia and in accordance with National Health and Medical Research Council (NHMRC) Australia guidelines.

Decision letter and Author response
Decision letter https://doi.org/10.7554/eLife.63452.sa1
Author response https://doi.org/10.7554/eLife.63452.sa2

## Additional files

### Supplementary files

• Supplementary file 1. Identification of human proteins that bind specifically to mouse Clec9A. Table 2: Glycan modification of Clec9A dimers. Table 3: Proteins that showed an enhanced association with Clec9A in the presence of RNF41. Table 4: Quantification of Clec9A and Ubiquitin ubiquitinated peptides.

• Transparent reporting form

### Data availability

The mass spectrometry proteomics data have been deposited to the ProteomeXchange Consortium via the PRIDE partner repository with the dataset identifier PXD018926. All further data and protocols supporting the current study are included in the article.

The following dataset was generated:

| Author(s) | Year | Dataset title | Dataset URL | Database and Identifier |
|---|---|---|---|---|
| Schittenhelm RB, Lahoud MH | 2020 | RNF41 regulation of the dendritic cell receptor Clec9A | https://www.ebi.ac.uk/pride/archive/projects/PXD018926 | PRIDE, PXD018926 |

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
