## [Decision Letter]

**Acceptance summary:**

This paper studies the regulation of the Clec9A receptor that is expressed on a subset of dendritic cells (DC) and contribute to cross-presentation, a key process for recognition of tumor antigens by T cells. The main finding of the paper is that Clec9A expression is regulated by an E3 ligase named RNF41, a previously described member of the RING family known to induce the degradation of several proteins via their ubiquitination. Steady state levels of Clec9A are regulated via their association with RNF4 and exposure of DC to dead cells induces a transient dissociation of RNF41 leading to increased levels of Clec9A and enhanced cross-presentation of antigens associated with dead cells.

**Decision letter after peer review:**

Thank you for submitting your article "RNF41 regulates the damage recognition receptor Clec9A and antigen cross-presentation in dendritic cells" for consideration by *eLife*. Your article has been reviewed by Satyajit Rath as the Senior Editor, a Reviewing Editor, and two reviewers. The following individual involved in review of your submission has agreed to reveal their identity: Philippe Benaroch (Reviewer #2).

The reviewers have discussed the reviews with one another and the Reviewing Editor has drafted this decision to help you prepare a revised submission.

Summary:

This paper studies the regulation of the Clec9A receptor that is expressed on a subset of dendritic cells (DC) and contribute to cross-presentation, a key process for recognition of tumor antigens by T cells. The main finding of the paper is that Clec9A expression is regulated by an E3 ligase named RNF41, a previously described member of the RING family known to induce the degradation of several proteins via their ubiquitination. Steady state levels of Clec9A are regulated via their association with RNF4 and exposure of DC to dead cells induces a transient dissociation of RNF41 leading to increased levels of Clec9A and enhanced cross-presentation of antigens associated with dead cells. The study encompasses a large number of well-designed and controlled experiments leading the two reviewers to raise a series of comments (see below). A major issue raised by both reviewers relates to the mechanisms through which the ectodomain of Clec9A can be subjected to ubiquitination by RNF4, including the cellular location of that process. However, considering that tackling this issue is very challenging they only require you to fairly state that this remains an important issue and present your thoughts on the matter.

Reviewer #1:

Tullett et al., study the regulation of the critical DC receptor Clec9A, finding that it is regulated by the E3 Ub ligase RN41 resulting in down regulating DC cross-presentation/cross priming of dead cell derived antigens to T cells The paper presents important information regarding cross-presentation, a process at the heart of T cell recognition of tumors and many viruses and other intracellular pathogens.

1) IMHO, by far the most interesting/important/intriguing finding in this study is the ubiquitylation of the Clec9A extracellular domain. How does this happen? Is the extracellular domain somehow transported to the cytoplasmic side of the containing membrane and then flipped back to the extracellular side? Is the Ub machinery present in the lumen? Both seem unlikely, with the former perhaps more so. Note that cell surface Ub was first reported by Irv Weissman's lab long ago (1986, in Science no less PMID:3003913). I don't think that this has ever been explained. I agreed to review the paper based on the Abstract describing cell surface Ub. Granted, there are plenty of other interesting findings in this paper. But, still, I am disappointed by the absence of addressing this intriguing finding, which could have enormous implications for cell biology.

2) Please explain the detergent extract experiment with more detail (Figure 2). How can GAPDH not be in either IGEPAL soluble or insoluble fractions? Where else can it be?

3) Figure 3. Clec9A associated proteins? Meaning what? I could not find the details of this experiment anywhere (perhaps my fault); in any event, an outline should be in the Results. How can PSMB1 show up here? Does this mean that Clec9A is exposed to the cytosol? Or is the proteasome in the lumen (a la Cresswell's last paper on proteasomes in class II compartments?). How can just one proteasome component show up? It isn't assembled (but aren't unassembled proteasome components degraded rapidly? The pathway analysis is uninformative/misleading, and should perhaps be in the supplemental data, if shown at all. Shouldn't this experiment be done as well in the presence of a proteasome inhibitor to detect interactions of Clec9A destined for degradation?

Reviewer #2:

The C-type lectin like receptor Clec9A (or DNGR1) is known to play a key role in the presentation by MHC I and II of dead cell derived antigens. The present study of Tullet et al., aimed at deciphering how Clec9A is regulated in dendritic cells. The authors found that Clec9A is regulated by an E3 ligase named RNF41, a previously described member of the RING family known to induce the degradation of several proteins via their ubiquitination. Levels of Clec9A are regulated via their association with RNF4 which takes place in steady state in DC, reducing the levels of Clec9A available. Importantly, exposure of DC to dead cells induces a transient dissociation of RNF41 leading to increased levels of Clec9A and in fine better cross presentation of antigens associated with dead cells.

The study represents a large body of experiments carefully controlled and achieved. Some of them are elegant. The manuscript is clearly written. The work presented deserves publication in *eLife* in principle because it brings several important new findings regarding a new level of regulation of the antigen presentation capacity of dendritic cells. I have however two main concerns that should be addressable and several minor comments to improve the study.

1) Among the several important findings reported in the study, the fact that the ectodomain of Clec9A is the substrate of RNF41 is well demonstrated but highly surprising. Like the authors, as far as I know this is the first example of Ubiquitination taking place on the intralumenal domain of a transmembrane protein. The lysines of the cytoplasmic and transmembrane domains of Clec9A were not modified by RNF41 while the ectodomain appears to be ubiquitinated at multiple sites. Since RNF41 is in all likelihood a cytosolic protein, how can it access the multiple lysines of the ectodomain of Clec9A? Where does the ubiquitination of Clec9A by RNF41 take place in the cell? The authors propose that it could take place at two locations either (i) in the ER or (ii) in the endosomes.

i) Regarding the first possibility, the authors propose that the association of Erlin1/2 with Clec9A they document by mass spec, reflects the involvement of the ERAD pathway, (Erlin1/2 have been indeed associated with the ERAD pathway in the literature). Thus, I guess they mean that Clec9A could be first retro-translocated, then ubiquitinated by RNF41 in the cytosol and targeted to the proteasome for degradation. This sounds an attractive explanation but it should be supported by additional data establishing a direct link with the ERAD pathway.

Moreover, the text should be clearer regarding the localization of RNF41 (Discussion). Do they authors consider that RNF41 could access to the lumen of the ER and how? In this matter, the ultra-structural analysis of RNF41 localization in DC presented in Figure 2—figure supplement 2 should be improved. The single immunogold RNF41 staining appears present at low density on multiple membranes which precise identity cannot be deduced only from the morphology of the membranes. Co-localization with ER markers (calnexin and calreticulin that seems to chaperone Clec9A) and endosomal markers would be required.

ii) Regarding the second hypothesis, the authors observe that association between Clec9A and RNF41 is favored at acidic pH. They suggest therefore that both proteins may interact in the lysosomes. Again, this should be directly supported by additional data either by EM as explained above, or by other means. And here also, the question of the access of RNF41 to the lumen of the endosome should be discussed, at least.

2) The authors use all along their study the cell line named MUTU DC but refer to it as cDC1, which might be misleading. It is important to extend a couple of the key findings to primary cDC1. I suggest 2 relatively simple experiments: The authors could measure the impact of primary cDC1 exposure to dead cells on the levels of expression of Clec9A at different time. They also can use their nice PLA assay to follow in cDC1, the association between the two proteins (Clec9A and RNF41) in the absence or presence of dead cells.

---

## [Author Response]

Summary:The reviewers have discussed the reviews with one another and the Reviewing Editor and their views concur. This paper studies the regulation of the Clec9A receptor that is expressed on a subset of dendritic cells (DC) and contribute to cross-presentation, a key process for recognition of tumor antigens by T cells. The main finding of the paper is that Clec9A expression is regulated by an E3 ligase named RNF41, a previously described member of the RING family known to induce the degradation of several proteins via their ubiquitination. Steady state levels of Clec9A are regulated via their association with RNF4 and exposure of DC to dead cells induces a transient dissociation of RNF41 leading to increased levels of Clec9A and enhanced cross-presentation of antigens associated with dead cells. The study encompasses a large number of well-designed and controlled experiments leading the two reviewers to raise a series of comments (see below). A major issue raised by both reviewers relates to the mechanisms through which the ectodomain of Clec9A can be subjected to ubiquitination by RNF4, including the cellular location of that process. However, considering that tackling this issue is very challenging they only require you to fairly state that this remains an important issue and present your thoughts on the matter.

We thank the Editors and reviewers for their recommendations. We have updated the manuscript to state that this remains an important issue and have expanded on the Discussion to reflect this.

Reviewer #1:Tullett et al., study the regulation of the critical DC receptor Clec9A, finding that it is regulated by the E3 Ub ligase RN41 resulting in down regulating DC cross-presentation/cross priming of dead cell derived antigens to T cells The paper presents important information regarding cross-presentation, a process at the heart of T cell recognition of tumors and many viruses and other intracellular pathogens.1) IMHO, by far the most interesting/important/intriguing finding in this study is the ubiquitylation of the Clec9A extracellular domain. How does this happen? Is the extracellular domain somehow transported to the cytoplasmic side of the containing membrane and then flipped back to the extracellular side? Is the Ub machinery present in the lumen? Both seem unlikely, with the former perhaps more so.

We agree that ubiquitination of the Clec9A extracellular domain is an intriguing finding. At this stage, we cannot definitively conclude whether the Clec9A extracellular domain is transported to the cytoplasmic side or whether ubiquitination occurs within endosomes. We would like to share new data with the reviewers, although we would prefer not to include this preliminary data in the manuscript at this time.

We hypothesised that transport of the Clec9A ectodomain to the cytosol would involve translocation of the Clec9A ectodomain through intracellular membranes, and that translocon component Sec61, or the AAA+ ATPase-VCP/p97, were likely to be involved in such translocation. We found that inhibition of Sec61 and VCP/p97, using the inhibitors ExoA and Eeyarestatin I (ES1) respectively, did not inhibit RNF41-mediated regulation of Clec9A levels (Author response image 1). Furthermore, inhibition of Sec61 did not appear to affect RNF41-mediated ubiquitination of Clec9A (Author response image 1). Thus, our data suggests that Sec61-mediated translocation is not necessary for RNF41-mediated ubiquitination of Clec9A, or its downstream processing for proteasomal degradation. Thus, while we cannot rule out translocation of the Clec9A ectodomain to the cytosol through alternative translocation mediators, this does not appear to be mediated via Sec61 or VCP/P97.

Alternatively, Professor Peter Cresswell and colleagues have recently demonstrated both ubiquitination and proteasomal machinery within endo-lysosomal compartments of the cell (Sengupta et al., 2019). We propose that this is a likely mechanism for RNF41-mediated regulation of internalised Clec9A receptors. Further research is required to determine the specific location of this interactions and whether it occurs on the cytoplasmic side of membranes or within endosomes. The Discussion has been expanded to address this issue in more detail.

**Author response image 1. sa2fig1:** RNF41-mediated regulation is not dependent on Sec61 or VCP/p97. (A) RNF41 mediated regulation of Clec9A levels is not dependent on Sec61 or VCP/p97. 293F cells were co-transfected with constructs encoding FLAG-tagged full-length mClec9A in the absence and presence of RNF41. At 15 hours, an aliquot of cells was harvested (left panel; prior to RNF41-mediated effects on Clec9A levels), and remaining cells were cultured for a further 6 h in the presence or absence of 20 μg/ml of ExoA or 10 μM ES1. Cellular lysates (input) were analyzed by WB. (B) RNF41 mediated ubiquitination of mClec9A is not dependent on Sec61. 293F cells were co-transfected with mClec9A-FLAG, Ub-Myc, and RNF41. At 15h, cells were cultured for 6 hours in the presence or absence of 12.5 μg/ml of ExoA. IP Clec9A complexes were analyzed for ubiquitination by WB using anti-Ubiquitin Ab. Cellular lysates (input) were analyzed by WB (representative of two experiments).

Note that cell surface Ub was first reported by Irv Weissman's lab long ago (1986, in Science no less PMID:3003913). I don't think that this has ever been explained. I agreed to review the paper based on the Abstract describing cell surface Ub. Granted, there are plenty of other interesting findings in this paper. But, still, I am disappointed by the absence of addressing this intriguing finding, which could have enormous implications for cell biology.

We thank the reviewer for this suggestion. We agree that this is an intriguing finding with significant implications, and have acknowledged the Weissman discovery in our Discussion.

2) Please explain the detergent extract experiment with more detail (Figure 2). How can GAPDH not be in either IGEPAL soluble or insoluble fractions? Where else can it be?

The detergent extractions with Digitonin and IGEPAL were performed sequentially, thus GAPDH is found in the Digitonin soluble fraction where most cytosolic proteins would be expected, rather than the IGEPAL soluble or resistant fractions. We have now modified the Figure 2 legend for clarity.

The updated Figure 2A legend now reads: **“**Cellular fractions were prepared from MutuDC 1940 by sequential detergent lysis and centrifugation conditions. In brief, DC were initially treated with digitonin and centrifugation, and digitonin soluble fractions harvested (containing GAPDH, EEA-1). Digitonin resistant fractions were then subjected to IGEPAL lysis and centrifugation, and IGEPAL soluble fractions (containing Calreticulin and Rab11) and IGEPAL resistant fractions harvested. Total lysates (Tx100) or cellular fractions (Digitonin soluble, Sol; IGEPAL soluble, Sol; IGEPAL resistant, Resistant) were analysed by Western blot (WB). Representative of 5 independent experiments.”

3) Figure 3. Clec9A associated proteins? Meaning what? I could not find the details of this experiment anywhere (perhaps my fault); in any event, an outline should be in the Results.

We thank the reviewer for bringing this to our attention. To identify novel Clec9A interactions that may facilitate Clec9A ectodomain access to ubiquitination and degradation machinery, we utilised cells transfected with Clec9A +/- RNF41. We performed Clec9A immunoprecipitations under higher stringency lysis conditions, and identified proteins that co-immunoprecipitated with Clec9A using LC/MS/MS. For clarity, we have replaced the phrase “Clec9A-associated proteins” with “Clec9A-interacting proteins” and have updated the manuscript to clearly describe the details of this experiment and include the outline in the Results and in the Figure legend.

The Results section now reads: “To elucidate the molecular machinery that facilitates access of Clec9A ectodomains to degradation machinery, we performed a proteomic analysis of Clec9A-interacting proteins in transfected cells. FLAG-tagged mClec9A was immunoprecipitated from cells transfected with Clec9A in the presence or absence of RNF41, digested with trypsin and subjected to LC-MS/MS analysis.”

How can PSMB1 show up here? Does this mean that Clec9A is exposed to the cytosol? Or is the proteasome in the lumen (a la Cresswell's last paper on proteasomes in class II compartments?).

Our studies revealed that immunoprecipitation of Clec9A from cells co-expressing Clec9A and RNF41 resulted in co-precipitation of PSMB1. This likely reflects the presence of K48-ubiquitinated Clec9A that is targeted for proteasomal degradation. At this stage, we cannot determine whether (1) Clec9A is exposed or translocated to the cytosol for degradation, or (2) Clec9A is exposed to ubiquitination and proteasomal machinery within endosomal compartments as proposed by Cresswell and colleagues (Sengupta et al., 2019). As outlined in the response to reviewer #1, point 1, we propose that ubiquitination of Clec9A in the endosome lumen is a likely mechanism for RNF41-mediated regulation of internalised Clec9A receptors but further studies will be required to determine the mechanism. This will be the subject of future studies, and the Discussion has been expanded to address this issue in more detail.

How can just one proteasome component show up? It isn't assembled (but aren't unassembled proteasome components degraded rapidly?

In order to identify novel Clec9A-interactions that may play a direct role in regulating Clec9A function, we utilised:

1) High stringency conditions for detergent lysis (1% Triton X-100, 150 mM NaCl, 1 mM EDTA) for co-IP of Clec9A interacting proteins.

2) Stringent criteria/ cut-offs for proteomic analysis: 4-fold or greater increase in the abundance of peptides (Clec9A interacting proteins) considering a false discovery rate (FDR) cutoff of 0.05, when comparing Clec9A-interacting proteins in the presence of transfected full-length RNF41 compared with a vector control (Figure 3—figure supplement 1F) or RNF41-∆ RING (Figure 3—figure supplement 1G).

We observed further proteasomal components if we adjusted our criteria. For example, PSMB5 was identified in IP of Clec9A at almost 2-fold increased abundance in the presence of RNF41 compared with RNF41-∆ RING, considering an FDR cutoff of 0.05. We expect that further proteasomal components would be identified if milder conditions were used for detergent lysis and Co-IP, or if the cells were pre-treated with proteasomal inhibitors (eg. MG132) to avoid rapid degradation of proteasomal components.

The pathway analysis is uninformative/misleading, and should perhaps be in the supplemental data, if shown at all.

We have moved the pathway analysis to the supplemental data (Figure 3—figure supplement 1H), as suggested by the reviewer. We have included this analysis as we believe the pathway analysis offers a visual demonstration of the types of Clec9A interacting proteins that are associated with the control of Clec9A ubiquitination and processing.

Shouldn't this experiment be done as well in the presence of a proteasome inhibitor to detect interactions of Clec9A destined for degradation?

This was indeed a question that we considered when we initiated these studies. However, our proteomic studies of immunoprecipitated Clec9A showed high sensitivity and depth of protein coverage, and enabled identification of ubiquitinated Clec9A, and Clec9A-interacting proteins, even in the absence of proteasomal inhibitors.

While the addition of proteasomal inhibitors, such as MG132, may increase the levels of proteins destined for degradation, we were concerned that the global effects on proteasomal degradation may reduce specificity of our screen for Clec9A-interacting proteins by increasing the abundance of indirect interactions. However, this is an area that we aim to follow up in the future.

Reviewer #2:The C-type lectin like receptor Clec9A (or DNGR1) is known to play a key role in the presentation by MHC I and II of dead cell derived antigens. The present study of Tullet et al., aimed at deciphering how Clec9A is regulated in dendritic cells. The authors found that Clec9A is regulated by an E3 ligase named RNF41, a previously described member of the RING family known to induce the degradation of several proteins via their ubiquitination. Levels of Clec9A are regulated via their association with RNF4 which takes place in steady state in DC, reducing the levels of Clec9A available. Importantly, exposure of DC to dead cells induces a transient dissociation of RNF41 leading to increased levels of Clec9A and in fine better cross presentation of antigens associated with dead cells.The study represents a large body of experiments carefully controlled and achieved. Some of them are elegant. The manuscript is clearly written. The work presented deserves publication in eLife in principle because it brings several important new findings regarding a new level of regulation of the antigen presentation capacity of dendritic cells. I have however two main concerns that should be addressable and several minor comments to improve the study.1) Among the several important findings reported in the study, the fact that the ectodomain of Clec9A is the substrate of RNF41 is well demonstrated but highly surprising. Like the authors, as far as I know this is the first example of Ubiquitination taking place on the intralumenal domain of a transmembrane protein. The lysines of the cytoplasmic and transmembrane domains of Clec9A were not modified by RNF41 while the ectodomain appears to be ubiquitinated at multiple sites. Since RNF41 is in all likelihood a cytosolic protein, how can it access the multiple lysines of the ectodomain of Clec9A? Where does the ubiquitination of Clec9A by RNF41 take place in the cell? The authors propose that it could take place at two locations either (i) in the ER or (ii) in the endosomes.i) Regarding the first possibility, the authors propose that the association of Erlin1/2 with Clec9A they document by mass spec, reflects the involvement of the ERAD pathway, (Erlin1/2 have been indeed associated with the ERAD pathway in the literature). Thus, I guess they mean that Clec9A could be first retro-translocated, then ubiquitinated by RNF41 in the cytosol and targeted to the proteasome for degradation. This sounds an attractive explanation but it should be supported by additional data establishing a direct link with the ERAD pathway.

Our data revealed that RNF41 can regulate newly synthesized Clec9A, as identified by less complex, high mannose glycosylation of Clec9A (Figure 3E, F and Figure 3—figure supplement 1D, E), indicative of regulation occurring at the ER. Furthermore, treatment with Brefeldin A, which inhibits transport of proteins from the ER/Golgi to the cell surface, did not inhibit RNF41 mediated degradation of newly synthesized Clec9A (Figure 3E,F). We propose that the Erlin 1/2 interaction lends further support to an ER-associated degradation pathway, and may play a role in facilitating access or retro-translocation of Clec9A to the cytosol where it can be ubiquitinated by RNF41. We agree that further additional data will be required to establish the direct link and have modified our Discussion to reflect this.

Moreover, the text should be clearer regarding the localization of RNF41 (lines 415-421). Do they authors consider that RNF41 could access to the lumen of the ER and how? In this matter, the ultra-structural analysis of RNF41 localization in DC presented in Figure 2—figure supplement 2 should be improved. The single immunogold RNF41 staining appears present at low density on multiple membranes which precise identity cannot be deduced only from the morphology of the membranes.

Our single RNF41 staining by immuno-EM indicates that RNF41 can associate with multiple intracellular membranes (Figure 2—figure supplement 2), consistent with the putative myristoylation sequence at position 2 of RNF41. We have used the morphology of the membranes as a guide to identification of the types of intracellular membranes (Zeuschner at al., 2006). We have updated the results to make this clearer. Furthermore, as the size of the Ab-protein A complex exceeds the thickness of the intracellular membrane, it is difficult at this stage to differentiate between whether the RNF41 detected is in the lumen or on the cytoplasmic side of the intracellular membranes. Further experimental approaches will be required to determine whether RNF41 can access the lumen of the ER or endosomal compartments. We have updated the Discussion to make this clearer.

Co-localization with ER markers (calnexin and calreticulin that seems to chaperone Clec9A) and endosomal markers would be required.

The association of RNF41 with endosomal membranes is further supported by our initial confocal microscopy co-localisation studies in DC of RNF41 with EEA1 and Rab11 using Flt3L-derived DC (Figure 2—figure supplement 1E). Previous studies have already described the association of RNF41 (Nrdp1) with ER markers including Calnexin, KDEL and PDI in alternative cell types using microscopy and fractionation studies (eg. Fry et al., 2011). Our biochemical fractionation studies (Figure 5—figure supplement 1G, H) support this data. However, we agree that further colocalization studies of endogenous RNF41 with ER and specific endosomal markers in cDC would further strengthen the understanding of RNF41 in the regulation of Clec9A and DC function. Unfortunately, we do not currently have a panel of markers and detection conditions that are compatible with our RNF41 Ab and PLA conditions to address this.

ii) Regarding the second hypothesis, the authors observe that association between Clec9A and RNF41 is favored at acidic pH. They suggest therefore that both proteins may interact in the lysosomes. Again, this should be directly supported by additional data either by EM as explained above, or by other means. And here also, the question of the access of RNF41 to the lumen of the endosome should be discussed, at least.

We thank the reviewer for their suggestion. We agree that additional data on colocalization would further our understanding of RNF41 regulation; Unfortunately, we do not currently have a compatible panel of markers and detection conditions to investigate this. We have expanded the Discussion to address the question of RNF41 access to the endosomal lumen.

2) The authors use all along their study the cell line named MUTU DC but refer to it as cDC1, which might be misleading. It is important to extend a couple of the key findings to primary cDC1. I suggest 2 relatively simple experiments: The authors could measure the impact of primary cDC1 exposure to dead cells on the levels of expression of Clec9A at different time. They also can use their nice PLA assay to follow in cDC1, the association between the two proteins (Clec9A and RNF41) in the absence or presence of dead cells.

We thank the reviewer for their suggestion. We have extended the Clec9A-RNF41 interaction studies to primary cDC1 in the presence and absence of dead cells, and demonstrate that the Clec9A-RNF41 interaction is decreased in the presence of dead cells at 4 hours, consistent with our findings using the cDC1 line MutuDC1940.

This data has been included in Figure 5—figure supplement 1C, and the Results updated as follows: “The degree of Clec9A-RNF41 interaction was significantly reduced in the presence of dead cells at 4 hours both in the cDC1 cell line MutuDC 1940 and in primary splenic cDC1 (Figure 5A and Figure 5—figure supplement 1C), although neither Clec9A nor RNF41 levels were reduced (Figure 5—figure supplement 1D).”.